# Computationally Efficient High-Dimensional Bayesian Optimization via Variable Selection

**Yihang Shen**[1]  **Carl Kingsford**[1]

[1]Computational Biology Department, School of Computer Science, Carnegie Mellon University

**Abstract**    Bayesian Optimization (BO) is a widely-used method for the global optimization of black-box functions. While BO has been successfully applied to many scenarios, scaling BO algorithms to high-dimensional domains remains a challenge. Optimizing such functions by vanilla BO is time-consuming. Alternative strategies for high-dimensional BO that are based on the idea of embedding the high-dimensional space to one with low dimensions are sensitive to the choice of the embedding dimension, which needs to be pre-specified. We develop a new computationally efficient high-dimensional BO method that leverages variable selection. We analyze the computational complexity of our algorithm and demonstrate its efficacy on several synthetic and real problems through empirical evaluations.

## 1 Introduction

We study the problem of globally maximizing a black-box function $f(\mathbf{x})$ with an input domain $\mathcal{X}$, where the function has certain properties: (1) Computation of its first and second-order derivatives is challenging, rendering gradient-based optimization algorithms unsuitable; (2) It is expensive to evaluate the function, hence some classical global optimization algorithms such as evolutionary algorithms (EA) are not applicable. These two properties are frequently encountered in real-world optimization problems, particularly in domains such as automated machine learning (Nickson et al., 2014) and robotics (Calandra et al., 2016).

Bayesian optimization (BO) is a popular global optimization method to solve the problem described above. It aims to obtain the input $\mathbf{x}^*$ that maximizes the function $f$ by iteratively selecting queries that are likely to achieve the maximum and evaluating the function on those queries. BO has been successfully applied in many scenarios such as hyper-parameter tuning (Snoek et al., 2012; Klein et al., 2017), automated machine learning (Nickson et al., 2014; Yao et al., 2018), reinforcement learning (Brochu et al., 2010; Marco et al., 2017; Wilson et al., 2014), robotics (Calandra et al., 2016; Berkenkamp et al., 2016), and chemical design (Griffiths and Hernández-Lobato, 2020; Negoescu et al., 2011; Griffiths et al., 2022). However, most problems described above that have been solved by BO successfully have black-box functions with low-dimensional domains, typically with $D \leq 20$ (Frazier, 2018). Scaling BO to high-dimensional black-box functions is challenging because (1) due to the curse of dimensionality, the global optima is harder to find as $D$ increases; and (2) computationally, vanilla BO is extremely time consuming on functions with large $D$. As the need for global optimization of high-dimensional black-box functions grows in fields such as algorithm configuration (Hutter et al., 2010), computer vision (Bergstra et al., 2013), and biology (Gonzalez et al., 2015), developing new BO algorithms that can effectively optimize such functions is crucial for practical applications.

A common approach for high-dimensional BO algorithms assumes that the black-box function has an effective subspace with dimension $d_e \ll D$ (Djolonga et al., 2013; Wang et al., 2016; Moriconi et al., 2020; Nayebi et al., 2019; Letham et al., 2020). To exploit this assumption, these algorithms first embed the high-dimensional search space $\mathcal{X}$ into a lower-dimensional space of dimension $d$, which is chosen by the user, and then perform vanilla BO in the embedding space to suggest new queries.

Subsequently, they project these queries back to the original space and evaluate the function $f$. These algorithms can be computationally efficient since BO is performed in a lower-dimensional space. It has been theoretically shown by Wang et al. (2016) that if the embedding dimension $d$ is greater than or equal to $d_e$, then the global optimum can be found by searching queries in the embedding space with probability one. However, the effective subspace dimension $d_e$ is usually unknown, and choosing a suitable $d$ can be difficult. Previous work, such as Eriksson and Jankowiak (2021), shows that the performance of embedding-based algorithms can be sensitive to the choice of $d$, and there is limited research on how to select $d$ heuristically. Letham et al. (2020) also points out that when projecting the optimal point from the embedding space back to the original space, there is no guarantee that the resulting point is within $\mathcal{X}$ and thus the algorithm may fail to find an optimum within the input domain, although this problem is partially addressed by mapping the input bounds from the original space to the embedding space in their work.

We develop a new algorithm, called VS-BO (Variable Selection Bayesian Optimization). Our method assumes that all the $D$ variables (elements) of the input $\mathbf{x}$ can be divided into two distinct sets $\mathbf{x} = \{\mathbf{x}_{ipt}, \mathbf{x}_{nipt}\}$, where $\mathbf{x}_{ipt}$ denotes important variables that have a significant impact on the output value of $f$, while $\mathbf{x}_{nipt}$ refers to unimportant variables with little or no effect on the output. Previous work such as Hutter et al. (2014) demonstrates that the performance of many machine learning methods is heavily influenced by only a small subset of hyperparameters, which justifies our assumption. We propose a robust strategy for identifying $\mathbf{x}_{ipt}$ and performing BO on the space of $\mathbf{x}_{ipt}$ to reduce computational time. Our method can automatically learn the dimension of $\mathbf{x}_{ipt}$, eliminating the need for pre-specifying the hyperparameter $d$ as in embedding-based algorithms. Since the space of $\mathbf{x}_{ipt}$ is axis-aligned, our method overcomes issues caused by space projection. We analyze the computational complexity of VS-BO, showing that our method can decrease the computational complexity of both fitting the Gaussian Process (GP) and optimizing the acquisition function. Our empirical results show that VS-BO performs well on synthetic and real problems. The source code of VS-BO is at https://github.com/Kingsford-Group/vsbo.

## 2  Related work

The basic framework of BO has two steps for each iteration: First, a GP model is used as the surrogate to model $f$ based on all the previous query-output pairs $\left(\mathbf{x}^{1:n}, y^{1:n}\right)$:

$$y^{1:n} \sim \mathcal{N}\left(\mathbf{0}, K(\mathbf{x}^{1:n}, \Theta) + \sigma_0^2 \mathbf{I}\right). \tag{1}$$

Here, $y^{1:n} = [y^1, \ldots, y^n]^\top$ is a $n$-dimensional vector, $y^i = f(\mathbf{x}^i) + \epsilon^i$ is the output of $f$ with random noise $\epsilon^i \sim \mathcal{N}(0, \sigma_0^2)$, and $K(\mathbf{x}^{1:n}, \Theta)$ is a $n \times n$ covariance matrix where its entry $K_{i,j} = k(\mathbf{x}^i, \mathbf{x}^j, \Theta)$ is the value of a kernel function $k$ in which $\mathbf{x}^i$ and $\mathbf{x}^j$ are the $i$-th and $j$-th queries respectively. $\Theta$ and $\sigma_0$ are hyper-parameters of GP that will be optimized each iteration, and $\mathbf{I}$ is the $n \times n$ identity matrix. We set the mean vector of (1) as a zero vector to implicitly assume that the observations have been standardized. A detailed description of the GP and its applications can be found in Williams and Rasmussen (2006).

Given a new input $\mathbf{x}$, we can compute the posterior distribution of $f(\mathbf{x})$ from the GP, which is again a Gaussian distribution with mean $\mu(\mathbf{x} \mid \mathbf{x}^{1:n}, y^{1:n})$ and variance $\sigma^2(\mathbf{x} \mid \mathbf{x}^{1:n})$ that have the following forms:

$$\mu(\mathbf{x} \mid \mathbf{x}^{1:n}, y^{1:n}) = \mathbf{k}(\mathbf{x}, \mathbf{x}^{1:n})[K(\mathbf{x}^{1:n}, \Theta) + \sigma_0^2 \mathbf{I}]^{-1} y^{1:n}, \tag{2}$$
$$\sigma^2(\mathbf{x} \mid \mathbf{x}^{1:n}) = k(\mathbf{x}, \mathbf{x}, \Theta) - \mathbf{k}(\mathbf{x}, \mathbf{x}^{1:n})[K(\mathbf{x}^{1:n}, \Theta) + \sigma_0^2 \mathbf{I}]^{-1} \mathbf{k}(\mathbf{x}, \mathbf{x}^{1:n})^\top.$$

Here, $\mathbf{k}(\mathbf{x}, \mathbf{x}^{1:n}) = [k(\mathbf{x}, \mathbf{x}^1, \Theta), \ldots, k(\mathbf{x}, \mathbf{x}^n, \Theta)]$ is a $n$-dimensional vector.

The second step of BO is to use $\mu$ and $\sigma$ to construct an acquisition function $acq$ and maximize it to get the new query $\mathbf{x}^{new}$, on which the function $f$ is evaluated to obtain the new pair $(\mathbf{x}^{new}, y^{new})$:

$$\mathbf{x}^{new} = \mathrm{argmax}_{\mathbf{x}\in\mathcal{X}} \; acq(\mu(\mathbf{x} \mid \mathbf{x}^{1:n}, y^{1:n}), \sigma(\mathbf{x} \mid \mathbf{x}^{1:n})). \tag{3}$$

A wide variety of methods have been proposed that are related to high-dimensional BO, and most of them are based on some extra assumptions on intrinsic structures of the domain $\mathcal{X}$ or the function $f$. As mentioned in the previous section, a considerable body of algorithms is based on the assumption that the black-box function has an effective subspace with a significantly smaller dimension than $\mathcal{X}$. Among them, REMBO (Wang et al., 2016) uses a randomly generated matrix as the projection operator to embed $\mathcal{X}$ to a low-dimensional subspace. SI-BO (Djolonga et al., 2013), DSA (Ulmasov et al., 2016) and MGPC-BO (Moriconi et al., 2020) propose different ways to learn the projection operator from data, of which the major shortcoming is that a large number of data points are required to make the learning process accurate. HeSBO (Nayebi et al., 2019) uses a hashing-based method to do subspace embedding. ALEBO (Letham et al., 2020) aims to improve the performance of REMBO with several novel refinements. There are also some methods that try to learn the subspace via Variational Autoencoders, such as Grosnit et al. (2021), Maus et al. (2022), and Notin et al. (2021).

Another assumption is that the black-box function has an additive structure. Kandasamy et al. (2015) first develops a high-dimensional BO algorithm called Add-GP by adopting this assumption. They derive a simplified acquisition function and prove that the regret bound is linearly dependent on the dimension. Their framework was subsequently generalized by Li et al. (2016); Wang et al. (2017) and Rolland et al. (2018).

Several approaches attempt to solve the high-dimensional BO problem by developing more efficient methods to optimize the acquisition function instead of adding extra assumptions. For example, Rana et al. (2017) builds a sequence of GPs and optimizes a series of acquisition functions to make the gradient-based methods applicable even in the region where the acquisition function is flattened, and Kirschner et al. (2019) develops a method called LineBO to optimize the acquisition function on a one-dimensional line each time.

Our method assumes that some variables are more important than others, which is an approach to axis-aligned subspace embedding. Several previous approaches propose different methods to choose axis-aligned subspaces in high-dimensional BO. Li et al. (2016) uses the idea of dropout, i.e, for each iteration of BO, a subset of variables is randomly chosen and optimized, while our work chooses variables that are important in place of the randomness. Gupta et al. (2020) proposes to optimize the acquisition function on a finite set of axis-aligned subspaces, while the dimension of the subspace is still pre-specified. Eriksson and Jankowiak (2021) develops a method called SAASBO, which uses the idea of Bayesian inference. SAASBO defines a prior distribution for each hyper-parameter in the kernel function $k$, and for each iteration the parameters are sampled from posterior distributions and used in the step of optimizing the acquisition function. Since those priors restrict parameters to concentrate near zero, the method is able to learn a sparse axis-aligned subspace (SAAS) during the BO process. The main drawback of SAASBO is that it is very time consuming. While traditionally it is assumed that the function $f$ is very expensive to evaluate so that the runtime of BO itself does not need to be considered, previous work such as Ulmasov et al. (2016) points out that in some applications the runtime of BO cannot be neglected. Spagnol et al. (2019) proposes a high-dimensional BO framework similar to our method. Their work uses Hilbert Schmidt Independence criterion (HSIC) to select variables and uses the chosen variables to do BO. However, they use simple heuristics to handle unimportant variables, which affects the performance of the method, and they do not analyze the computational complexity of their method. Also, they do not provide a comprehensive comparison with other high-dimensional BO methods: their method is only compared with the method in Li et al. (2016) on several synthetic functions.

**Algorithm 1** VS-BO

---

1: **Input:** black-box function $f(\mathbf{x})$ with no analytic form, $\mathcal{X} = [0, 1]^D$, $N_{init}$, $N$, $N_{vs}$
2: **Output:** $\mathbf{x}^{max}$
3: Initialize the set of $\mathbf{x}_{ipt}$ to be all variables in $\mathbf{x}$, $\mathbf{x}_{ipt} = \mathbf{x}$, and $\mathbf{x}_{nipt} = \emptyset$
4: Uniformly sample $N_{init}$ points $\mathbf{x}^i$ and evaluate $y^i = f(\mathbf{x}^i) + \epsilon^i$, let $\mathcal{D} = \{(\mathbf{x}^i, y^i)\}_{i=1}^{N_{init}}$
5: Initialize multivariate Gaussian distribution $p(\mathbf{x} \mid \mathcal{D})$
6: **for** $t = N_{init} + 1, N_{init} + 2, \ldots N_{init} + N$ **do**
7:     **if** $\mathrm{mod}(t - N_{init}, N_{vs}) = 0$ **then**
8:         Variable selection to update $\mathbf{x}_{ipt}$ and let $\mathbf{x}_{nipt} = \mathbf{x} \setminus \mathbf{x}_{ipt}$ (Algorithm 2)
9:         Update $p(\mathbf{x} \mid \mathcal{D})$, then derive the conditional distribution $p(\mathbf{x}_{nipt} \mid \mathbf{x}_{ipt}, \mathcal{D})$
10:     **end if**
11:     Fit a GP to $\mathcal{D}_{ipt} := \{(\mathbf{x}_{ipt}^i, y^i)\}_{i=1}^{t-1}$
12:     Maximize the acquisition function to obtain $\mathbf{x}_{ipt}^t$.
13:     Sample $\mathbf{x}_{nipt}^t$ from $p(\mathbf{x}_{nipt} \mid \mathbf{x}_{ipt}^t, \mathcal{D})$
14:     Evaluate $y^t = f(\mathbf{x}^t) + \epsilon^t = f(\{\mathbf{x}_{ipt}^t, \mathbf{x}_{nipt}^t\}) + \epsilon^t$ and update $\mathcal{D} = \mathcal{D} \cup \{(\mathbf{x}^t, y^t)\}$
15: **end for**
16: **return** $\mathbf{x}^{max}$ which is equal to $\mathbf{x}^i$ with maximal $y^i$

---

## 3 Framework of VS-BO

Given the black-box function $f(\mathbf{x}) : \mathcal{X} \to \mathbb{R}$ in the domain $\mathcal{X}$, without loss of generality, we assume $\mathcal{X} = [0, 1]^D$ with a large dimension $D$, the goal of high-dimensional BO is to find the maximizer $\mathbf{x}^* = \mathrm{argmax}_{\mathbf{x} \in \mathcal{X}} f(\mathbf{x})$ efficiently. VS-BO assumes that all variables in $\mathbf{x}$ can be divided into important variables $\mathbf{x}_{ipt}$ and unimportant variables $\mathbf{x}_{nipt}$. Our algorithm uses different strategies to select new queries for variables from two different sets. For variables that are deemed important, we use vanilla BO (or other BO frameworks such as TuRBO (Eriksson et al., 2019)) to select queries, while for variables that are considered unimportant, we use a sampling strategy to select queries.

    The framework of VS-BO is described in Algorithm 1. For every $N_{vs}$ iterations VS-BO will update $\mathbf{x}_{ipt}$ and $\mathbf{x}_{nipt}$ (line 8 in Algorithm 1), and for every BO iteration $t$ only variables in $\mathbf{x}_{ipt}$ are used to fit the GP (line 11 in Algorithm 1), and the new query of important variables $\mathbf{x}_{ipt}^t$ is obtained by maximizing the acquisition function (line 12 in Algorithm 1). VS-BO learns a multivariate Gaussian distribution $p(\mathbf{x} \mid \mathcal{D})$ from the existing query-output pairs $\mathcal{D}$ (line 5, 9 in Algorithm 1). Ideally, $p(\mathbf{x} \mid \mathcal{D})$ will assign a higher probability to $\mathbf{x}$ that has a corresponding higher $y$ value. Once $\mathbf{x}_{ipt}^t$ is obtained, the algorithm samples $\mathbf{x}_{nipt}^t$ from the conditional distribution $p(\mathbf{x}_{nipt} \mid \mathbf{x}_{ipt}^t, \mathcal{D})$ (line 13 in Algorithm 1), concatenates it with $\mathbf{x}_{ipt}^t$ and evaluates $f(\{\mathbf{x}_{ipt}^t, \mathbf{x}_{nipt}^t\})$.

    We have two novel contributions that distinguish our method from Spagnol et al. (2019). First, we introduce a novel variable selection method that fully leverages the information contained within the fitted GP model. Secondly, we enhance the BO framework by integrating an evolutionary algorithm to enable a more precise sampling of unimportant variables. The subsequent subsections provide detailed explanations of these two aspects.

### 3.1 Variable selection

The variable selection step in VS-BO (Algorithm 2) can be separated into two substeps: (1) calculate the importance score (*IS*) of each variable (line 3 in Algorithm 2), and (2) do the stepwise-forward variable selection (Derksen and Keselman, 1992) according to the importance scores.

    For step one, we extended a gradient-based *IS* calculation method, called Grad-IS, based on Paananen et al. (2019). Intuitively, if the partial derivative of the function $f$ with respect to one variable is large on average, then the variable ought to be important. Since the derivative of $f$ is

---

**Algorithm 2** Variable Selection (line 8 in Algorithm 1)

---

1: **Input**: $\mathcal{D} = \{(\mathbf{x}^i, y^i)\}_{i=1}^t$, $r_{stop}$
2: **Output**: Set of important variables $\mathbf{x}_{ipt}$
3: Fit a GP to $\mathcal{D}$ and calculate importance scores $IS$, where $IS[j]$ is the score of the $j$-th variable
4: Sort variables according to their importance scores from the most important to the least, $[\mathbf{x}_{s(1)}, \ldots, \mathbf{x}_{s(D)}]$
5: **for** $m = 1, 2, \ldots, D$ **do**
6:     Fit a GP to $\mathcal{D}_m := \{(\mathbf{x}^i_{s(1):s(m)}, y^i)\}_{i=1}^{t-1}$ where $\mathbf{x}^i_{s(1):s(m)}$ is the $i$-th input with only the first $m$ important variables. Let $L_m$ to be the value of final negative marginal log likelihood
7:     **if** $m \geq 3$ and $L_{m-1} - L_m \leq \max\{0, (L_{m-2} - L_{m-1})/r_{stop}\}$ **then**
8:         **break**
9:     **end if**
10: **end for**
11: **return** $\mathbf{x}_{ipt} = \{\mathbf{x}_{s(1)}, \ldots, \mathbf{x}_{s(m-1)}\}$

---

unknown, VS-BO instead estimates the expectation of the gradient of the posterior mean from a fitted GP model, normalized by the posterior standard deviation:

$$IS = \mathbb{E}_{\mathbf{x} \sim Unif(\mathcal{X})} \left[ \left| \frac{\nabla_{\mathbf{x}} \mathbb{E}_{p(f(\mathbf{x})|\mathbf{x}, \mathcal{D})}\left[f(\mathbf{x})\right]}{\sqrt{Var_{p(f(\mathbf{x})|\mathbf{x}, \mathcal{D})}\left[f(\mathbf{x})\right]}} \right| \right] = \mathbb{E}_{\mathbf{x} \sim Unif(\mathcal{X})} \left[ \left| \frac{\nabla_{\mathbf{x}} \mu(\mathbf{x} \mid \mathcal{D})}{\sigma(\mathbf{x} \mid \mathcal{D})} \right| \right] \tag{4}$$

$$\approx \frac{1}{N_{is}} \sum_{k=1}^{N_{is}} \left| \frac{\nabla_{\mathbf{x}} \mu(\mathbf{x}^{(k)} \mid \mathcal{D})}{\sigma(\mathbf{x}^{(k)} \mid \mathcal{D})} \right| \quad \mathbf{x}^{(k)} \overset{i.i.d}{\sim} Unif(\mathcal{X}). \tag{5}$$

Here $N_{is}$ represents the number of Monte Carlo samples for estimating $IS$, and both $\nabla_{\mathbf{x}} \mu(\cdot \mid \mathcal{D})$ and $\sigma(\cdot \mid \mathcal{D})$ have explicit forms. Both the Grad-IS and Kullback-Leibler Divergence (KLD)-based methods in Paananen et al. (2019) are estimations of $\mathbb{E}_{\mathbf{x} \sim Unif(\mathcal{X})} \left[ \left| \frac{\nabla_{\mathbf{x}} \mathbb{E}_{p(f(\mathbf{x})|\mathbf{x}, \mathcal{D})}\left[f(\mathbf{x})\right]}{\sqrt{Var_{p(f(\mathbf{x})|\mathbf{x}, \mathcal{D})}\left[f(\mathbf{x})\right]}} \right| \right]$. Since the KLD method only calculates approximate derivatives around the chosen points in $\mathcal{D}$ that are always unevenly distributed, it is a biased estimator, while our importance score estimation is unbiased.

Each time the algorithm fits the GP to the existing query-output pairs, the marginal log likelihood (MLL) of GP is maximized by updating parameters $\Theta$ and $\sigma_0$. VS-BO takes negative MLL as the loss and uses its value as the stopping criteria of the stepwise-forward selection. More specifically, VS-BO sequentially selects variables according to the importance score, and when a new variable is added, the algorithm will fit the GP again by only using those chosen variables and records a new loss (line 6 in Algorithm 2). If the loss converges, then the selection step stops (line 7 in Algorithm 2) and all those already chosen variables are important variables. Stepwise-forward selection can adaptively determine the number of variables to include in the model. On the one hand, it tends to select a small number of variables from $\mathbf{x}$ when only a few of them are truly important; On the other hand, in the worst-case scenario where every variable is equally important, stepwise-forward selection may end up selecting almost all the variables, effectively reducing VS-BO to vanilla BO.

In our experiments, we implement a modified version called VS-momentum to achieve a more robust variable selection process. The basic idea is to leverage the queries obtained after each variable selection step to gain additional insights into the accuracy of the selected variables. We say that the variable selection at iteration $t + N_{vs}$ is in an accurate case when $\max_{k \in \{t+1, \ldots, t+N_{vs}\}} y^k > \max_{k \in \{1, \ldots, t\}} y^k$, otherwise it is in an inaccurate case. In the accurate case, VS-BO first uses a recursive feature elimination (RFE) algorithm to remove redundant variables in $\mathbf{x}_{ipt}$ that are selected at $t$, then it adds new variables into the remaining only if the loss decreases evidently

(Figure 3a in the appendix). In the inaccurate case, variables selected at $t$ will not be considered at $t + N_{vs}$ unless they still obtain very high importance scores at $t + N_{vs}$ (marked by the blue box in Figure 3b). New variables are added via the stepwise-forward algorithm. Algorithm 3 in the appendix provides the pseudo-code of VS-momentum.

## 3.2 Sampling for unimportant variables

To obtain the new value of unimportant variables for each iteration, we propose a method based on the Covariance Matrix Adaptation Evolution Strategy (CMA-ES). CMA-ES is an evolutionary algorithm for numerically optimizing a function. For each generation $k$, the algorithm samples new offsprings from a multivariate Gaussian distribution $\mathcal{N}\left(m^{(k-1)}, (\sigma^{(k-1)})^2\right)$ and updates $m^{(k-1)}$ and $(\sigma^{(k-1)})^2$ based on these new samples and their corresponding function values. Details of this algorithm can be seen in Hansen (2016).

Using the same approach as CMA-ES, VS-BO uses the initialized data $\{(\mathbf{x}^i, y^i)\}_{i=1}^{N_{init}}$ to initialize the multivariate Gaussian distribution $p(\mathbf{x} \mid \mathcal{D})$ (line 5 in Algorithm 1), and for every $N_{vs}$ iterations, it updates the distribution based on new query-output pairs (line 9 in Algorithm 1). Because of the properties of Gaussian distributions, the conditional distribution $p(\mathbf{x}_{nipt} \mid \mathbf{x}_{ipt}, \mathcal{D})$ is easily derived and is also a multivariate Gaussian distribution. Therefore, $\mathbf{x}_{nipt}^t$ can be sampled from the Gaussian distribution $p(\mathbf{x}_{nipt} \mid \mathbf{x}_{ipt}^t, \mathcal{D})$ (line 13 in Algorithm 1) when $\mathbf{x}_{ipt}^t$ is obtained.

Compared to BO, it is much faster to update the evolutionary algorithm and obtain new queries, although these queries are less precise than those from BO. VS-BO takes advantage of the strength of these two methods by using them on different variables. Important variables are crucial to the function value, therefore VS-BO uses the framework of BO on them to obtain precise queries. Unimportant variables do not affect the function value too much so there is no need to spend a large time budget to search for extremely precise values. Hence, they are determined by CMA-ES to reduce runtime. In the case when the variable selection step is imprecise, VS-BO resembles the CMA-ES algorithm rather than random sampling. Hence, incorporating this sampling strategy can enhance the overall algorithm's robustness.

## 4 Computational complexity analysis

From the theoretical perspective, we prove that running BO by only using those important variables is able to decrease the runtime of both the step of fitting the GP and maximizing the acquisition function. Specifically, we have the following proposition:

**Proposition 4.1.** *Suppose the dimension of $\mathbf{x}_{ipt}$ is $p$ and the Quasi-Newton method (QN) is used for both fitting the GP and maximizing the acquisition function. Assume we choose the squared exponential kernel or the Matérn kernel as the kernel function, and upper confidence bound (UCB) or expected improvement (EI) as the acquisition function. If only variables in $\mathbf{x}_{ipt}$ are used for fitting the GP and maximizing the acquisition function, then the complexity of each step of QN is $\mathcal{O}(p^2 + pn^2 + n^3)$ for fitting the GP and $\mathcal{O}(p^2 + pn + n^2)$ for maximizing the acquisition function, where $n$ is the number of queries that are already obtained.*

The proof is in section B of the appendix. Note that the method for fitting the GP and maximizing the acquisition function under the framework of BoTorch (Balandat et al., 2020), a python library for BO, is limited-memory BFGS, which is indeed a QN method.

For vanilla BO, the computational complexity of each step of QN is $\mathcal{O}(D^2 + Dn^2 + n^3)$ for fitting the GP and $\mathcal{O}(D^2 + Dn + n^2)$ for maximizing the acquisition function. However, in the case of VS-BO, the large dimension value $D$ is replaced with $p$ by selecting a smaller subset of variables. This reduction in dimensionality can significantly decrease the runtime of BO. Empirical evidence in Figure 4 of the appendix shows that compared to vanilla BO, VS-BO can significantly reduce the runtime required for both fitting the GP and optimizing the acquisition function.

We also derive the computational complexity of the variable selection step, resulting in the following proposition:

**Proposition 4.2.** *Suppose the dimension of $\mathbf{x}$ is $D$, the dimension of $\mathbf{x}_{ipt}$ is $p$ and the Quasi-Newton method (QN) is used for fitting the GP. Assuming the squared exponential kernel or the Matérn kernel, the complexity of computing importance scores IS (line 3 of Algorithm 2) is $\mathcal{O}\left(Dn^2 + n^3 + N_{is}(Dn + n^2)\right)$, and the complexity of each step of QN for fitting the GP in each iteration of the stepwise forward selection (line 6 of Algorithm 2) is $\mathcal{O}(p^2 + pn^2 + n^3)$, where $n$ is the number of queries that are already obtained.*

The proof is also in section B of the appendix. The final component of the importance score computation complexity, $N_{is}(Dn + n^2)$, arises from the Monte Carlo sampling process (Eq. 5). However, this can be accelerated through parallelism, resulting in a rapid computation of *IS* in practice. When $p$ is small, the stepwise forward selection method is also not time-consuming since it typically requires only about $p$ iterations within each variable selection step. Moreover, we perform variable selection only once every $N_{vs}$ BO iterations ($N_{vs} = 20$ in our experiments).

## 5 Experiments

We compare VS-BO with a wide range of existing methods: vanilla BO, REMBO and its variant REMBO Interleave, HeSBO and ALEBO, LineBO and SAASBO. The details of implementations of these methods as well as hyperparameter settings are described in section C of the appendix. However, it should be noted that LineBO requires multiple evaluations of the black box function with multiple queries to determine the direction of the one-dimensional line at each iteration, making the comparison with other BO methods unfair. In our experiments, we evaluated the black box function with 10 different queries for each iteration of LineBO.

### 5.1 Synthetic problems

We use the Hartmann6 ($d_e = 6$), Styblinski-Tang4 ($d_e = 4$) and Branin ($d_e = 2$) functions as test functions. Previous high-dimensional BO studies extended these functions to high dimensions by adding unrelated variables. However, in our work, we presented a more challenging test setting by including both unrelated and unimportant (but not completely unrelated) variables. For example, with the standard Hartmann6 function $f_{Hartmann6}(\mathbf{x}_{[1:6]})$ we first construct a new function $F_{hm6}(\mathbf{x})$ by adding variables with importance weights $[1, 0.1, 0.01]$, $F_{hm6}(\mathbf{x}) = f_{Hartmann6}(\mathbf{x}_{[1:6]}) + 0.1f_{Hartmann6}(\mathbf{x}_{[7:12]}) + 0.01f_{Hartmann6}(\mathbf{x}_{[13:18]})$, and we further extend it to $D = 50$ by adding unrelated variables. For full details, please refer to section C. The dimension of the effective subspace of $F_{hm6}$ is 18, while the dimension of important variables is only 6. For each existing embedding-based method, we evaluate both $d = 4$ and $d = 6$.

Figure 1 shows the performance of VS-BO and other BO methods on the three synthetic functions. In the fixed iteration budget scenario (Figure 1a), the value on average found by VS-BO after 200 iterations is the best or comparable to the best in all three cases. When the wall clock time or CPU time budget is fixed (Figures 1b,c), VS-BO can achieve a high function value with high computational efficiency.

Vanilla BO under the framework of BoTorch can also achieve good performance for the fixed iteration budget, but it is computationally inefficient. For embedding-based methods, the results reflect some of their limitations. Firstly, the performance of these methods is more variable than that of VS-BO; for example, HeSBO with $d = 6$ performs very well in the Styblinski-Tang4 case but not in the others; Secondly, embedding-based methods are sensitive to the choice of the embedding dimension $d$: they perform especially poorly when $d$ is smaller than the dimension of important variables (see results of the Hartmann6 case) and may still perform poorly even when $d$ is larger (such as ALEBO with $d = 6$ in the Styblinski-Tang4 case), whereas VS-BO can automatically

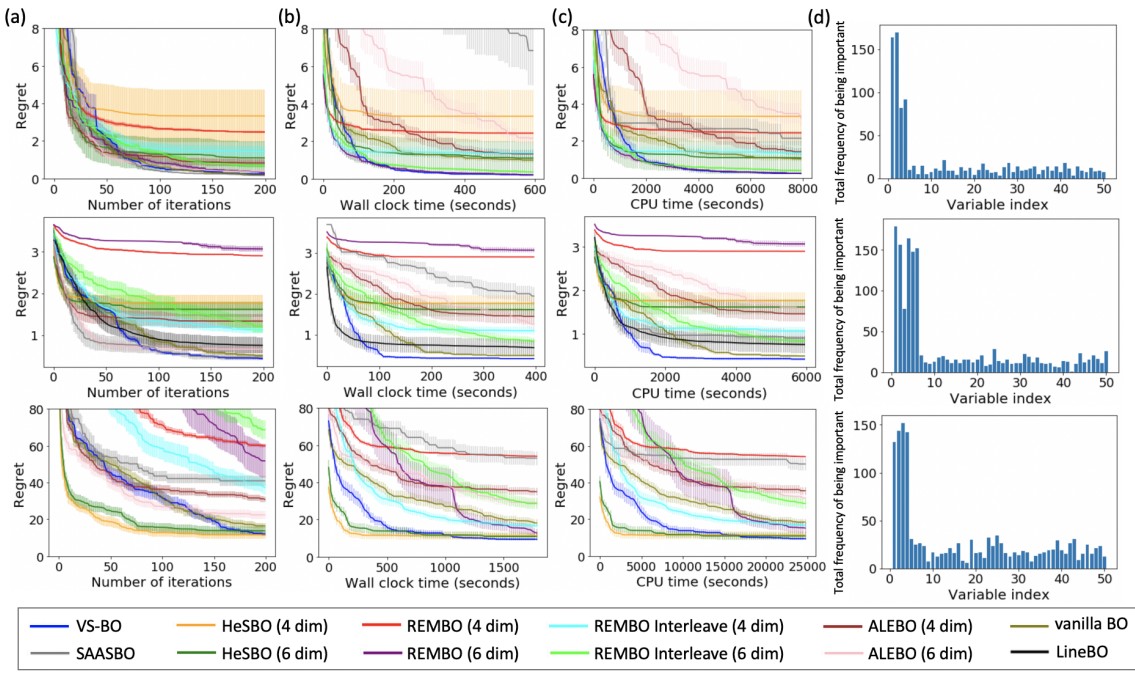

Figure 1: (a,b,c) Performance of BO methods on the Branin (first row), Hartmann6 (second row) and Styblinski-Tang4 (third row) test functions. For each test function, we perform 20 independent runs for each method except SAASBO which is very time-consuming, we perform 10 runs instead. We plot the mean and standard error of the regret (the maximal possible value - the current maximal value found) versus (a) iterations, (b) wall clock time or (c) CPU time. (d) The total frequency of being chosen as important for each variable. For Branin, the first two variables are the most important, for Hartmann6, the first six variables are the most important, while for Styblinski-Tang4, the first four variables are the most important.

learn the dimension. One advantage of embedding-based methods is that they may have a better performance than VS-BO within a very limited iteration budget (for example 50 iterations), which is expected since several data points are required for VS-BO to make accurate variable selection. Under the fixed iteration budget, SAASBO performs very well except in Styblinski-Tang4 case. However, SAASBO requires a significantly higher time budget for each iteration than other methods. For example, SAASBO requires approximately 9 hours to complete 200 iterations on the Branin case, whereas VS-BO can accomplish 500 iterations on this case in around 20 minutes. Section D shows results on VS-BO with larger numbers of iterations. Although LineBO evaluates the function 10 times for each iteration, it is still significantly worse than any other methods (Figure 5) except in the Hartmann6 case.

Figures 1d shows the frequency of being chosen as important for each variable in steps of variable selection of VS-BO. Since 20 runs of VS-BO are performed on each test function, each run has 200 iterations, and important variables are re-selected every 20 iterations, the maximum frequency that each variable can be chosen as important is 200. VS-BO can accurately identify important variables and simultaneously control false-positive selections. Figure 6 shows the number of variables chosen as important, i.e. the size of $\mathbf{x}_{ipt}$, at each variable selection step during the iterations on each test function. It is evident that only a small number of variables are deemed important in each step, and this number is close to the number of important variables in reality.

In contrast to the Branin case, the Harmann6 and Styblinski-Tang4 cases exhibit a different pattern, where the total frequency of selection for secondary important variables (variables 7-12 in

the Harmann6 case and variables 5-8 in the Styblinski-Tang4 case) is not significantly higher than that of unrelated variables. We speculate that this may be due to the small importance weights of these secondary important variables, which have a negligible impact on the function value. To test this hypothesis, we modify the importance weights of the test functions from $[1, 0.1, 0.01]$ to $[1, 0.5, 0.1]$ and $[1, 0.9, 0.8]$, and evaluate VS-BO on these new test functions. Figure 7 shows that when the importance weights are increased to $[1, 0.5, 0.1]$, VS-BO successfully selects both the most important variables and the secondary important variables, and Figure 8 shows that when the importance weights are increased to $[1, 0.9, 0.8]$, VS-BO successfully selects all the related variables. These results provide further evidence of the robustness of the variable selection module in VS-BO.

We conduct a study to assess the robustness of two hyper-parameters in VS-BO: $r_{stop}$ and $N_{is}$. The former controls the point at which variable selection stops (line 7 in Algorithm 2), while the latter represents the number of Monte Carlo samples used to estimate the importance score $IS$ (Equation (5)). We test different values of $r_{stop}$ (5, 10, 50, and 100) and $N_{is}$ (1,000, 10,000, 50,000, and 100,000) on both the Branin and Hartmann6 cases, and the results in Figure 10 indicate that both hyper-parameters are highly robust across different values.

Spagnol et al. (2019) proposed several simple heuristics for sampling unimportant variables, and we compare our sampling strategy, which involves sampling from the CMA-ES posterior, with their heuristics by creating a variant of VS-BO called VS-BO-mix. All components of VS-BO-mix remained the same as those in VS-BO, except for the sampling strategy, which was replaced with the best heuristic from Spagnol et al. (2019), called the mix strategy. This mix strategy involves sampling values of unimportant variables from a uniform distribution with probability 0.5 and using the previous query's values that have the highest function value with probability 0.5. We compare the performance of VS-BO and VS-BO-mix on three synthetic functions, and the results, depicted in Figure 11, demonstrate that our sampling strategy outperforms the mix strategy.

To investigate the impact of the momentum mechanism on the performance of VS-BO, we compare it with a version of VS-BO without momentum, denoted as VS-BO-nomom. As shown in Figure 12, although the regret curves of VS-BO-nomom are highly similar to those of VS-BO (Figure 12a), VS-BO-nomom appears to be less effective in selecting secondary important variables, particularly in the Branin case (Figure 12b). We also replace the sampling strategy of VS-BO-nomom with the mix strategy (denoted as VS-BO-nomom-mix) and compare it with both VS-BO and VS-BO-nomom. The performance of VS-BO-nomom-mix is inferior to that of both VS-BO and VS-BO-nomom, which further indicates that our sampling strategy outperforms the mix strategy.

## 5.2 Real-world problems

We compare VS-BO with other methods on two real-world problems. First, VS-BO is tested on the rover trajectory optimization problem presented in Wang et al. (2017), a problem with a 60-dimensional input domain. Second, it is tested on the vehicle design problem MOPTA08 (Jones, 2008), a problem with 124 dimensions. On these two problems, we evaluate both $d = 6$ and $d = 10$ for each embedding-based method, except we omit ALEBO with $d = 10$ since it is very time consuming. The detailed settings of these two problems are described in section C of the appendix.

Figures 2a,b show the performance of VS-BO and other methods on these two problems. When the iteration budget is fixed (Figure 2a), SAASBO has the best performance, while Figure 2b shows that VS-BO is more computationally efficient than the other methods. Figure 2c shows the frequency of being chosen as important for each variable by VS-BO. As there is no ground truth of important variables in these real-world cases, we conduct a sampling experiment to test whether those more frequently-chosen variables are more important. Specifically, we generate a set of input points, for each point, we randomly sample the values of the first 5 variables from the input domain that have been chosen most frequently by our variable selection module and keep the values of other variables fixed with the values in the best query, the query that has the highest function value. We then calculate the function values for these inputs. We repeat this process for the first five variables

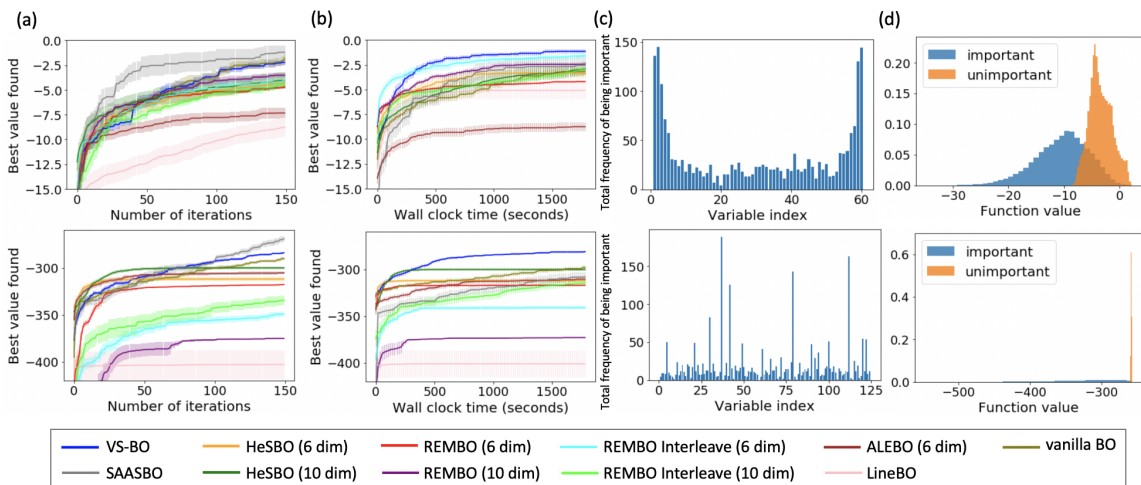

Figure 2: (a,b) Performance of BO methods on the rover trajectory (first row) and MOPTA08 (second row) problem. For each test function, we do 15 independent runs for each method except SAASBO which is very time consuming, we do 10 runs instead. We plot the mean and standard error of the best maximum value found by (a) iterations and (b) wall clock time. (c) The total frequency of being chosen as important for each variable. (d) The distribution of function values when sampling the first 5 variables that have been chosen most frequently (important) or the first 5 variables that have been chosen least frequently (unimportant) with all the other variables fixed.

that have been chosen least frequently, and evaluate the functions accordingly. Figure 2d shows that the variance of function values from the first set of input points is significantly higher than that from the second set, particularly on the MOPTA08 problem. Moreover, the mean of function values from the second set is significantly higher than that from the first set, which is attributed to the fact that the input points in the second set have their more frequently chosen variables fixed to the values obtained from the best query, which are approximately optimal values. These findings suggest that frequently selected variables have a more significant effect on the function value.

## 6 Conclusion

We present a novel approach, named VS-BO, for high-dimensional BO. Our method is based on the assumption that input variables can be partitioned into two categories: important and unimportant. We design different strategies to assign values to the identified important and unimportant variables, which is a crucial step toward enhancing the computational efficiency of VS-BO. Our experiments show the good performance of VS-BO, making it a valuable tool for optimizing high-dimensional black-box functions.

We also find some limitations of our method. First, when the dimension of the input increases, it becomes harder to do variable selection accurately. Therefore, embedding-based methods might still be the first choice when the input of a function has thousands of dimensions. It might be interesting to develop new algorithms that can do variable selection robustly even when the dimension is extremely large[1]. Further, Grad-IS might be invalid when variables are discrete or categorical, therefore new methods for calculating the importance score of these kinds of variables are needed. These are several directions for future improvements of VS-BO.

---

[1]After the preprint of our work (Shen and Kingsford, 2021) was released, Song et al. (2022) published a follow-up work for high-dimensional BO via variable selection; they use Monto Carlo tree search to select variables. Papenmeier et al. (2022) also published a high-dimensional BO that is related. We do some comparisons in section F.

## 7 Broader Impact Statement

We present a novel optimization approach that can be applied to black-box functions in a variety of fields, such as machine learning, computational chemistry, and bioinformatics. By speeding up the optimization process, our approach has the potential to accelerate scientific research, which will provide positive benefits to society.

The authors believe that this work presents no notable negative impacts to society or the environment.

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

## A  Variable selection with momentum mechanism

This section presents the VS-momentum mechanism in Figure 3, and the pseudo-code for this mechanism is provided in Algorithm 3.

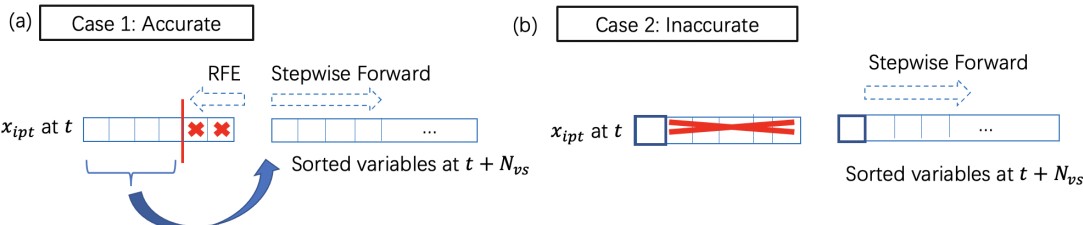

Figure 3: Momentum mechanism in VS-BO. (a) Accurate case, RFE is first used to remove redundant variables, and then new variables are added. (b) Inaccurate case, most variables are removed except those that are considered very important in both variable selection steps (blue box). New variables are then added.

---

**Algorithm 3** Variable Selection (VS) with Momentum

---

1: **Input**: Iteration index $t$, $\mathcal{D} = \{(\mathbf{x}^i, y^i)\}_{i=1}^t$, $N_{init}$, $N_{vs}$, $\hat{\mathbf{x}}_{ipt}$ which is the set of important variables chosen at iteration $t - N_{vs}$
2: **Output**: $\mathbf{x}_{ipt}$ which is the set of important variables chosen at this iteration
3: **if** $t = N_{init} + N_{vs}$ or $\hat{\mathbf{x}}_{ipt} = \mathbf{x}$ **then**
4:     **return** Algorithm 2
5: **else if** $\max_{k \in \{t-N_{vs}+1, t-N_{vs}+2, \ldots, t\}} y^k \leq \max_{k \in \{1, \ldots, t-N_{vs}\}} y^k$ **then**            ▷ Inaccurate case
6:     **return** Algorithm 4
7: **else**                                                                                     ▷ Accurate case
8:     **return** Algorithm 5
9: **end if**

---

---
**Algorithm 4** Momentum in the inaccurate case
---
1: **Input:** $\mathcal{D} = \{(\mathbf{x}^i, y^i)\}_{i=1}^t$, $N_{vs}$, $\hat{\mathbf{x}}_{ipt}$ which is the set of important variables chosen at iteration $t - N_{vs}$, $r_{stop}$
2: **Output:** $\mathbf{x}_{ipt}$ which is the set of important variables chosen at this iteration
3: Fit a GP to $\mathcal{D}$ and calculate important scores of variables $IS$ where $IS[j]$ is the important score of the j-th variable
4: Sort variables according to their important scores from the most important to the least, $[\mathbf{x}_{s(1)}, \ldots, \mathbf{x}_{s(D)}]$
5: **for** $n = 1, \ldots, D$ **do**
6:     **if** $\mathbf{x}_{s(n)} \notin \hat{\mathbf{x}}_{ipt}$ **then**
7:         **break**
8:     **end if**
9: **end for**
10: **for** $m = n, n + 1, \ldots, D$ **do**
11:     Fit a GP to $\mathcal{D}_m := \{(\mathbf{x}^i_{s(1):s(m)}, y^i)\}_{i=1}^{t-1}$ where $\mathbf{x}^i_{s(1):s(m)}$ is the $i$-th input with only the first $m$ important variables. Let $L_m$ to be the value of final negative marginal log likelihood
12:     **if** $m - n \geq 2$ and $L_{m-1} - L_m \leq \max\{0, (L_{m-2} - L_{m-1})/r_{stop}\}$ **then**
13:         **break**
14:     **end if**
15: **end for**
16: **return** $\mathbf{x}_{ipt} = \{\mathbf{x}_{s(1)}, \ldots, \mathbf{x}_{s(m-1)}\}$
---

---

**Algorithm 5** Momentum in accurate case

---

1: **Input:** $\mathcal{D} = \{(\mathbf{x}^i, y^i)\}_{i=1}^t$, $N_{vs}$, $\hat{\mathbf{x}}_{ipt}$ which is the set of important variables chosen at iteration $t - N_{vs}$. Let the cardinality of $\hat{\mathbf{x}}_{ipt}$ be $w$, $w = |\hat{\mathbf{x}}_{ipt}|$, $r_{stop}$
2: **Output:** $\mathbf{x}_{ipt}$ which is the set of important variables chosen at this iteration
3: Fit a GP to $\mathcal{D}$ and calculate important scores of variables $IS$ where $IS[j]$ is the important score of the j-th variable
4: Sort variables according to $IS$, $[\mathbf{x}_{s(1)}, \ldots, \mathbf{x}_{s(D)}]$, from the most important to the least
5: Fit a GP by using variables in $\hat{\mathbf{x}}_{ipt}$, i.e. fit a GP to $\{(\hat{\mathbf{x}}_{ipt}^i, y^i)\}_{i=1}^t$, and calculate important scores of these variables $\widehat{IS}$. Let $\hat{L}_w$ be the value of final negative marginal log likelihood
6: Sort variables in $\hat{\mathbf{x}}_{ipt}$ according to $\widehat{IS}$, $[\mathbf{x}_{s'(1)}, \ldots, \mathbf{x}_{s'(w)}]$, from the most important to the least.
7: **for** $m = w - 1, w - 2, \ldots, 0$ **do**                          ▷ Recursive feature elimination
8:     **if** $m = 0$ **then**
9:         Set $\mathbf{x}_{ipt} = \{\mathbf{x}_{s'(1)}\}$
10:         **break**
11:     **end if**
12:     Fit a GP by only using the first $m$ important variables according to $\widehat{IS}$. Let $\hat{L}_m$ to be the value of final negative marginal log likelihood
13:     **if** $\hat{L}_m > \hat{L}_{m+1}$ **then**
14:         Set $\mathbf{x}_{ipt} = \{\mathbf{x}_{s'(1)}, \ldots, \mathbf{x}_{s'(m+1)}\}$
15:         Set $L_0 = \hat{L}_{m+1}$
16:         **break**
17:     **end if**
18: **end for**
19: **for** $m = 1, 2, \ldots, D$ **do**                          ▷ Stepwise forward selection
20:     **if** $\mathbf{x}_{s(m)} \in \mathbf{x}_{ipt}$ **then**
21:         Set $L_m = L_{m-1}, L_{m-1} = L_{m-2}$
22:         **continue**
23:     **end if**
24:     Fit a GP by using variables in $\mathbf{x}_{ipt} \cup \{\mathbf{x}_{s(m)}\}$. Let $L_m$ to be the value of final negative marginal log likelihood
25:     **if** $m \geq 3$ and $L_{m-1} - L_m \leq \max\{0, (L_{m-2} - L_{m-1})/r_{stop}\}$ **then**
26:         **break**
27:     **end if**
28:     $\mathbf{x}_{ipt} = \mathbf{x}_{ipt} \cup \{\mathbf{x}_{s(m)}\}$
29: **end for**
30: **return** $\mathbf{x}_{ipt}$

---

## B Proof of Propositions related to computational complexity analysis

### B.1 Proof of Proposition 4.1

*Proof of Proposition 4.1.* Given query-output pairs $\mathcal{D} = \{(\mathbf{x}^i, y^i)\}_{i=1}^n$, the marginal log likelihood (MLL) that needs to be maximized at the step of fitting a GP has the following explicit form:

$$\log p(\Theta = \{\rho_{1:D}^2, \alpha_0^2\}, \sigma_0 \mid \mathcal{D}) = -\frac{1}{2}\mathbf{y}^\top M^{-1}\mathbf{y} - \frac{1}{2}\log|M| - \frac{n \log 2\pi}{2},$$

where $\mathbf{y} = [y^1, \ldots y^n]^\top$ is an $n$-dimensional vector, $M = \left(K(\mathbf{x}^{1:n}, \Theta) + \sigma_0^2\mathbf{I}\right)$, $\alpha_0^2$ is the signal amplitude of the kernel function, and $\rho_{1:D}^2$ are the inverse lengthscales of the kernel function. When the quasi-Newton method is used for maximizing MLL, the gradient should be calculated for each iteration:

$$\nabla_{\Theta, \sigma_0} \log p(\Theta, \sigma_0 \mid \mathcal{D}) = -\frac{1}{2}\mathbf{y}^\top M^{-1} \left(\nabla_{\Theta, \sigma_0}M\right) M^{-1}\mathbf{y} - \frac{1}{2}\mathrm{tr}\left(M^{-1} \left(\nabla_{\Theta, \sigma_0}M\right)\right).$$

When only variables in $\mathbf{x}_{ipt}$ are used, we define the distance between two queries $\mathbf{x}^i$ and $\mathbf{x}^j$ as:

$$d(\mathbf{x}^i, \mathbf{x}^j) = \sqrt{\sum_{m:m \in \mathbf{x}_{ipt}} \rho_m^2 (\mathbf{x}_m^i - \mathbf{x}_m^j)^2},$$

and all the other inverse squared length scales corresponding to unimportant variables are fixed to 0. Commonly chosen kernel functions are actually functions of the distance defined above, for example the squared exponential (SE) kernel is as the following:

$$k_{SE}(\mathbf{x}^i, \mathbf{x}^j, \Theta) = \alpha_0^2 \exp\left(-\frac{1}{2}d^2(\mathbf{x}^i, \mathbf{x}^j)\right),$$

and the Matérn-5/2 kernel is as the following:

$$k_{Mt}(\mathbf{x}^i, \mathbf{x}^j, \Theta) = \alpha_0^2 \left(1 + \sqrt{5}d(\mathbf{x}^i, \mathbf{x}^j) + \frac{5}{3}d^2(\mathbf{x}^i, \mathbf{x}^j)\right) \exp\left(-\sqrt{5}d(\mathbf{x}^i, \mathbf{x}^j)\right).$$

Since the cardinality of $\mathbf{x}_{ipt}$ is $p$, the cardinality of parameters in the kernel function that are not fixed to 0 is $p + 1$, hence the complexity of calculating the gradient of the distance is $\mathcal{O}(p)$. Therefore whatever using SE kernel or Matérn-5/2 kernel, the complexity of calculating $\nabla_\Theta k(\mathbf{x}^i, \mathbf{x}^j, \Theta)$ is $\mathcal{O}(p)$.

Since $M$ is a $n \times n$ matrix and each entry $M_{ij}$ equals to $k(\mathbf{x}^i, \mathbf{x}^j, \Theta) + \sigma_0^2 \mathbb{1}(i = j)$, the complexity of calculating $\nabla_{\Theta, \sigma_0}M$ is $\mathcal{O}(pn^2)$. The complexity of calculating the inverse matrix $M^{-1}$ is $\mathcal{O}(n^3)$ in general, and the following matrix multiplication and trace calculation need $\mathcal{O}(pn^2)$, therefore the complexity of calculating the gradient of MLL is $\mathcal{O}(pn^2 + n^3)$. Once the gradient is obtained, each quasi-Newton step needs additional $\mathcal{O}(p^2)$ time, therefore the complexity of one step of quasi-Newton method when fitting a GP is $\mathcal{O}(p^2 + pn^2 + n^3)$.

As described in section 2, the acquisition function is a function that depends on the posterior mean $\mu$ and the posterior standard deviation $\sigma$, hence the gradients of $\mu$ and $\sigma$ should be calculated when the gradient of the acquisition function is needed.

When only variables in $\mathbf{x}_{ipt}$ are used, the gradient of $\mu$ with respect to $\mathbf{x}_{ipt}$ has the following form:

$$\nabla_{\mathbf{x}_{ipt}} \mu(\mathbf{x}_{ipt} \mid \mathcal{D}) = \left(\nabla_{\mathbf{x}_{ipt}}\mathbf{k}(\mathbf{x}_{ipt}, \mathbf{x}_{ipt}^{1:n})\right) \left(K(\mathbf{x}_{ipt}^{1:n}, \Theta) + \sigma_0^2\mathbf{I}\right)^{-1} \mathbf{y}.$$

Here $\left(K(\mathbf{x}_{ipt}^{1:n}, \Theta) + \sigma_0^2 \mathbf{I}\right)^{-1} \mathbf{y}$ is fixed so that its value can be calculated in advance and stored as a $n$-dimensional vector. $\mathbf{k}(\mathbf{x}_{ipt}, \mathbf{x}_{ipt}^{1:n})$ is a $n$-dimensional vector of which each element is a kernel value between $\mathbf{x}_{ipt}$ and $\mathbf{x}_{ipt}^i$, hence the complexity of calculating the gradient of each element in $\mathbf{k}(\mathbf{x}_{ipt}, \mathbf{x}_{ipt}^{1:n})$ is $\mathcal{O}(p)$. Therefore, the complexity is $\mathcal{O}(pn)$ to calculate $\nabla_{\mathbf{x}_{ipt}} \mathbf{k}(\mathbf{x}_{ipt}, \mathbf{x}_{ipt}^{1:n})$ and $\mathcal{O}(pn)$ for additional matrix manipulation, hence the total complexity for calculating $\nabla_{\mathbf{x}_{ipt}} \mu(\mathbf{x}_{ipt} \mid \mathcal{D})$ is $\mathcal{O}(pn)$.

The gradient of $\sigma$ has the following form:

$$\nabla_{\mathbf{x}_{ipt}} \sigma(\mathbf{x}_{ipt} \mid \mathcal{D}) = \nabla_{\mathbf{x}_{ipt}} \sqrt{k(\mathbf{x}_{ipt}, \mathbf{x}_{ipt}, \Theta) - \mathbf{k}(\mathbf{x}_{ipt}, \mathbf{x}_{ipt}^{1:n})[K(\mathbf{x}_{ipt}^{1:n}, \Theta) + \sigma_0^2 \mathbf{I}]^{-1} \mathbf{k}(\mathbf{x}_{ipt}, \mathbf{x}_{ipt}^{1:n})^\top}$$

$$= -\frac{\left(\nabla_{\mathbf{x}_{ipt}} \mathbf{k}(\mathbf{x}_{ipt}, \mathbf{x}_{ipt}^{1:n})\right) \left(K(\mathbf{x}_{ipt}^{1:n}, \Theta) + \sigma_0^2 \mathbf{I}\right)^{-1}}{\sqrt{k(\mathbf{x}_{ipt}, \mathbf{x}_{ipt}, \Theta) - \mathbf{k}(\mathbf{x}_{ipt}, \mathbf{x}_{ipt}^{1:n})[K(\mathbf{x}_{ipt}^{1:n}, \Theta) + \sigma_0^2 \mathbf{I}]^{-1} \mathbf{k}(\mathbf{x}_{ipt}, \mathbf{x}_{ipt}^{1:n})^\top}}.$$

Once $\nabla_{\mathbf{x}_{ipt}} \mathbf{k}(\mathbf{x}_{ipt}, \mathbf{x}_{ipt}^{1:n})$ is calculated, $\mathcal{O}(pn + n^2)$ is needed for additional matrix manipulation, hence the total complexity for calculating $\nabla_{\mathbf{x}_{ipt}} \sigma(\mathbf{x}_{ipt} \mid \mathcal{D})$ is $\mathcal{O}(pn + n^2)$.

For commonly used acquisition functions such as upper confidence bound (UCB) (Auer, 2002):

$$UCB(\mathbf{x}_{ipt} \mid \mathcal{D}) = \mu(\mathbf{x}_{ipt} \mid \mathcal{D}) + \sqrt{\beta_n} \sigma(\mathbf{x}_{ipt} \mid \mathcal{D}),$$

and expected improvement (EI) (Močkus, 1975):

$$EI(\mathbf{x}_{ipt} \mid \mathcal{D}) = \left(\mu(\mathbf{x}_{ipt} \mid \mathcal{D}) - y_n^*\right) \Phi\left(\frac{\mu(\mathbf{x}_{ipt} \mid \mathcal{D}) - y_n^*}{\sigma(\mathbf{x}_{ipt} \mid \mathcal{D})}\right) + \sigma(\mathbf{x}_{ipt} \mid \mathcal{D}) \varphi\left(\frac{\mu(\mathbf{x}_{ipt} \mid \mathcal{D}) - y_n^*}{\sigma(\mathbf{x}_{ipt} \mid \mathcal{D})}\right),$$

where $y_n^* = \max_{i \le n} y^i$, $\Phi(\cdot)$ is the cumulative distribution function of the standard normal distribution, and $\varphi(\cdot)$ is the probability density function, once the gradients of $\mu$ and $\sigma$ are derived, only additional $\mathcal{O}(p)$ time is needed for vector calculation. Hence the total complexity of calculating the gradient of the acquisition function is $\mathcal{O}(pn + n^2)$. Again, once the gradient is obtained, each quasi-Newton step needs additional $\mathcal{O}(p^2)$, therefore the complexity of one step of quasi-Newton method for maximising the acquisition function is $\mathcal{O}(p^2 + pn + n^2)$.

$\square$

## B.2 Proof of Proposition 4.2

*Proof of Proposition 4.2.* We first prove the complexity of computing the importance scores *IS*. Since we use the following form to compute *IS*:

$$IS \approx \frac{1}{N_{is}} \sum_{k=1}^{N_{is}} \frac{\nabla_{\mathbf{x}} \mu(\mathbf{x}^{(k)} \mid \mathcal{D})}{\sigma(\mathbf{x}^{(k)} \mid \mathcal{D})} \quad \mathbf{x}^{(k)} \overset{i.i.d}{\sim} Unif(\mathcal{X}),$$

for each sampled point $\mathbf{x}^{(k)}$, we need to compute $\nabla_{\mathbf{x}} \mu(\mathbf{x}^{(k)} \mid \mathcal{D})$ and $\sigma(\mathbf{x}^{(k)} \mid \mathcal{D})$. The proof is similar to the proof of Proposition 4.1, except that now we need to use all $D$ dimensions of $\mathbf{x}$ rather than the dimensions of $\mathbf{x}_{ipt}$. $\nabla_{\mathbf{x}} \mu(\mathbf{x}^{(k)} \mid \mathcal{D})$ has the following form:

$$\nabla_{\mathbf{x}} \mu(\mathbf{x}^{(k)} \mid \mathcal{D}) = \left(\nabla_{\mathbf{x}^{(k)}} \mathbf{k}(\mathbf{x}^{(k)}, \mathbf{x}^{1:n})\right) \left(K(\mathbf{x}^{1:n}, \Theta) + \sigma_0^2 \mathbf{I}\right)^{-1} \mathbf{y}.$$

Here $\left(K(\mathbf{x}^{1:n}, \Theta) + \sigma_0^2 \mathbf{I}\right)^{-1} \mathbf{y}$ is a constant for different samples $\mathbf{x}^{(k)}$, we can therefore use $\mathcal{O}(Dn^2 + n^3)$ to compute it in advance ($\mathcal{O}(Dn^2)$ for computing $K(\mathbf{x}^{1:n}, \Theta) + \sigma_0^2 \mathbf{I}$ and $\mathcal{O}(n^3)$ for the matrix inversion).

The complexity of calculating the gradient of each element in $\mathbf{k}(\mathbf{x}^{(k)}, \mathbf{x}^{1:n})$ is $\mathcal{O}(D)$, hence the complexity is $\mathcal{O}(Dn)$ to calculate $\nabla_{\mathbf{x}^{(k)}} \mathbf{k}(\mathbf{x}^{(k)}, \mathbf{x}^{1:n})$ and the additional $\mathcal{O}(Dn)$ is needed for the matrix manipulation to compute $\nabla_{\mathbf{x}} \mu(\mathbf{x}^{(k)} \mid \mathcal{D})$. Therefore, once $\left(K(\mathbf{x}^{1:n}, \Theta) + \sigma_0^2 \mathbf{I}\right)^{-1} \mathbf{y}$ has been computed, the total complexity for computing $\nabla_{\mathbf{x}} \mu(\mathbf{x}^{(k)} \mid \mathcal{D})$ for each $\mathbf{x}^{(k)}$ is $\mathcal{O}(Dn)$.

$\sigma(\mathbf{x}^{(k)} \mid \mathcal{D})$ has the following form:

$$\sigma(\mathbf{x}^{(k)} \mid \mathcal{D}) = k(\mathbf{x}^{(k)}, \mathbf{x}^{(k)}, \Theta) - \mathbf{k}(\mathbf{x}^{(k)}, \mathbf{x}^{1:n}) [K(\mathbf{x}^{1:n}, \Theta) + \sigma_0^2 \mathbf{I}]^{-1} \mathbf{k}(\mathbf{x}^{(k)}, \mathbf{x}^{1:n})^\top.$$

Since $[K(\mathbf{x}^{1:n}, \Theta) + \sigma_0^2 \mathbf{I}]^{-1}$ is already computed, $\mathcal{O}(Dn + n^2)$ is needed for additional matrix manipulation, hence the total complexity for calculating $\sigma(\mathbf{x}^{(k)} \mid \mathcal{D})$ for each $\mathbf{x}^{(k)}$ is $\mathcal{O}(Dn + n^2)$.

Since $N_{is}$ points will be sampled, the total complexity for computing $IS$ is $\mathcal{O}(Dn^2 + n^3 + N_{is}(Dn + n^2))$.

As we proved in Proposition 4.1, if only $m$ sub-dimensions of $\mathbf{x}$ are used for fitting the GP (line 6 of Algorithm 2), the complexity of each step of the Quasi-Newton method is $\mathcal{O}(m^2 + mn^2 + n^3)$. Since we assume the dimension of $\mathbf{x}_{ipt}$ is $p$, we have $m = \mathcal{O}(p)$ in each iteration of the stepwise forward selection. Therefore, the complexity of each step of QN for fitting the GP in each iteration of the stepwise forward selection is $\mathcal{O}(p^2 + pn^2 + n^3)$, and we will only do around $p$ iterations of the stepwise forward selection within one variable selection step.

$\square$

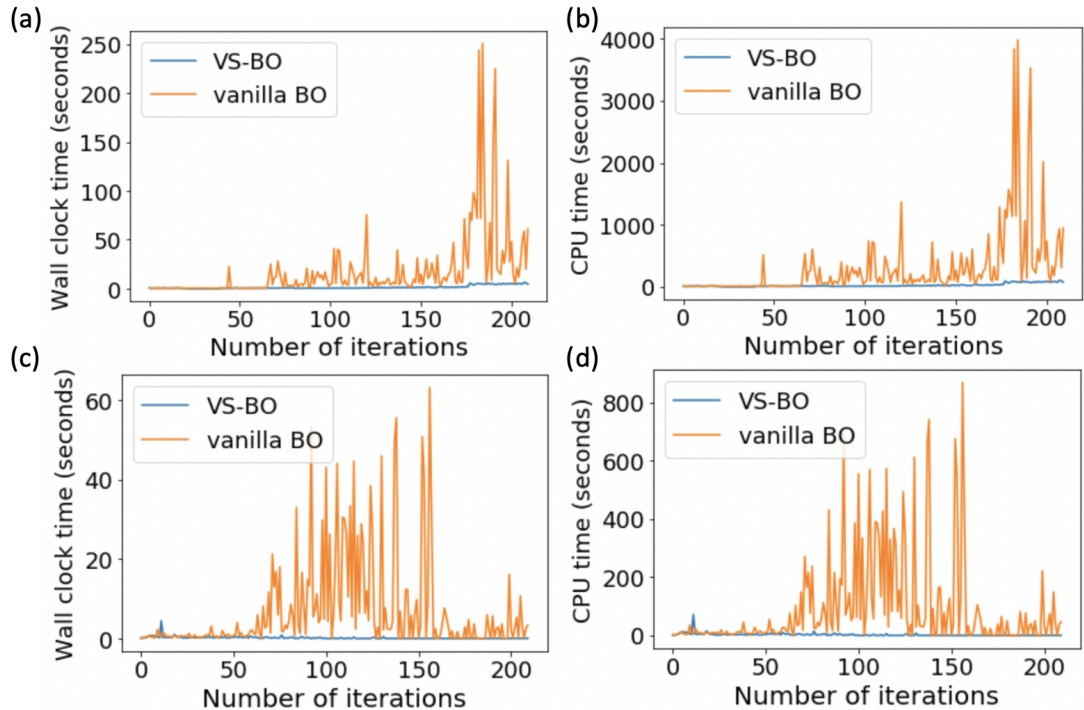

Figure 4: The wall clock time or CPU time comparison between VS-BO and vanilla BO for each iteration. The Branin test function with $d_e = [2, 2, 2]$ and $D = 50$ is run here. (a) Wall clock time comparison at the GP fitting step. (b) CPU time comparison at the GP fitting step. (c) Wall clock time comparison at the acquisition function optimization step. (d) CPU time comparison at the acquisition function optimization step.

## C  Detailed experimental settings and extended discussion of experimental results

We use the framework of BoTorch to implement VS-BO. We compare VS-BO to the following existing BO methods: vanilla BO, which is implemented by the standard BoTorch framework[2]; REMBO and its variant REMBO Interleave (Wang et al., 2016), of which the implementations are based on Metzen (2016)[3]; HeSBO (Nayebi et al., 2019) which has already been implemented in Adaptive Experimentation Platform (Ax)[4]; ALEBO[5] (Letham et al., 2020); DescentLineBO[6] (Kirschner et al., 2019); and SAASBO[7] (Eriksson and Jankowiak, 2021). At the time when we write this manuscript, the source codes of the approach in Spagnol et al. (2019) have not been released, so we cannot compare VS-BO with it. Both VS-BO and vanilla BO use Matérn 5/2 as the kernel function and expected improvement as the acquisition function, and use limited-memory BFGS (L-BFGS) (Liu and Nocedal, 1989) to fit GP and optimize the acquisition function. The number of initialized samples $N_{init}$ is set to 5 for all methods, and $N_{vs}$ in VS-BO is set to 20, $r_{stop}$ is set to 10, and $N_{is}$ is set to 10000 for all experiments. The number of the interleaved cycle for REMBO Interleave is set to 4. Since our algorithm aims to maximize the black-box function, all the test functions that have minimum points are converted to the corresponding negative forms. As described in section 5, in order to decide the direction of the one-dimensional line, for each iteration DescentLineBO needs to

---

[2]https://botorch.org
[3]https://github.com/jmetzen/bayesian_optimization
[4]https://github.com/facebook/Ax/tree/master/ax/modelbridge/strategies
[5]https://github.com/facebookresearch/alebo
[6]https://github.com/kirschnj/LineBO
[7]https://github.com/martinjankowiak/saasbo

evaluate the black box function multiple times with multiple queries (Algorithm 4 in Kirschner et al. (2019)), making the comparison between this method and other BO methods unfair. Because of this property, LineBO is actually not suitable for optimizing a function that is expensive to evaluate. In our experiments, DescentLineBO will evaluate the black box function with 10 different queries for each iteration while all the other methods evaluate once.

In synthetic experiments, as described in section 5.1, for each test function we add some unimportant variables as well as unrelated variables to make it high-dimensional. The standard Branin function $f_{Branin}$ has two dimensions with the input domain $\mathcal{X}_{Branin} = [-5, 10] \times [0, 10]$, and we construct a new Branin function $F_{branin}$ as the following:

$$F_{branin}(\mathbf{x}) = f_{Branin}(\mathbf{x}_{[1:2]}) + 0.1 f_{Branin}(\mathbf{x}_{[3:4]}) + 0.01 f_{Branin}(\mathbf{x}_{[5:6]}),$$

$$\mathbf{x} \in \left( \bigotimes_{i=1}^{3} \mathcal{X}_{Branin} \right) \bigotimes_{i=1}^{44} [0, 1]$$

where $\otimes$ represents the direct product. We use $d_e = [2, 2, 2]$ to represent the dimension of the effective subspace of $F_{branin}$, the total effective dimension is 6, however, the number of important variables is only 2.

Likewise, for the standard Hartmann6 function $f_{Hartmann6}$ that has six dimensions with the input domain $[0, 1]^6$, we construct $F_{hm6}$ as:

$$F_{hm6}(\mathbf{x}) = f_{Hartmann6}(\mathbf{x}_{[1:6]}) + 0.1 f_{Hartmann6}(\mathbf{x}_{[7:12]}) + 0.01 f_{Hartmann6}(\mathbf{x}_{[13:18]}) \quad \mathbf{x} \in [0, 1]^{50}$$

and use $d_e = [6, 6, 6]$ to represent the dimension of the effective subspace. For the Styblinski-Tang4 function $f_{ST4}$ that has four dimensions with the input domain $[-5, 5]^4$, we construct $F_{ST4}$ as:

$$F_{ST4}(\mathbf{x}) = f_{ST4}(\mathbf{x}_{[1:4]}) + 0.1 f_{ST4}(\mathbf{x}_{[5:8]}) + 0.01 f_{ST4}(\mathbf{x}_{[9:12]}) \quad \mathbf{x} \in [-5, 5]^{50}$$

and use $d_e = [4, 4, 4]$ to represent the dimension of the effective subspace. All synthetic experiments are run on the same Linux cluster that has 40 3.0 GHz 10-Core Intel Xeon E5-2690 v2 CPUs.

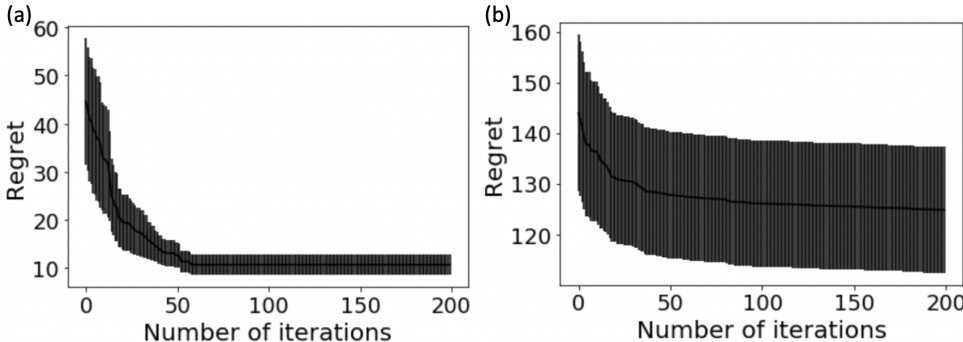

Figure 5: Performance of LineBO on the Branin (a) and Styblinski-Tang4 (b) function. We do 20 independent runs. We plot the mean and standard error of the regret versus iterations. Compared to Figure 1, we can see that the performance of LineBO is significantly worse than any other method on these two cases.

For real-world problems, the rover trajectory problem is a high-dimensional optimization problem with input domain $[0, 1]^{60}$. The problem setting in our experiment is the same as that in Wang et al. (2017).[8] MOPTA08 is another high-dimensional optimization problem with input domain

---

[8]The source code of this problem can be found in https://github.com/zi-w/Ensemble-Bayesian-Optimization.

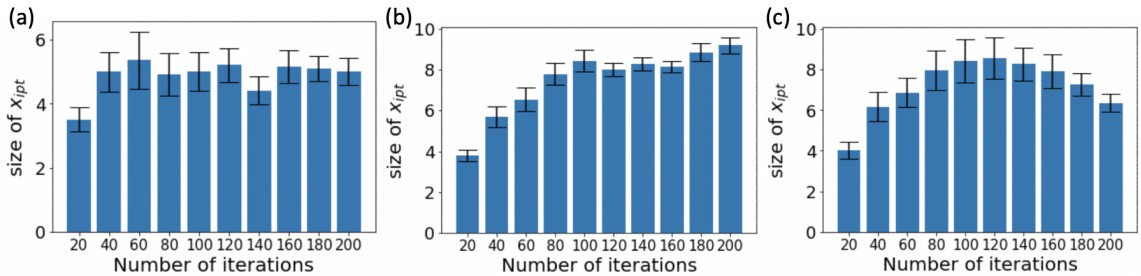

Figure 6: Number of variables chosen as important, i.e. $|\mathbf{x}_{ipt}|$, versus iterations on the Branin (a), Hartmann6 (b) and Styblinski-Tang4 (c) cases. We do 20 independent runs. We plot the mean and standard error of $|\mathbf{x}_{ipt}|$ for each variable selection step along the iterations.

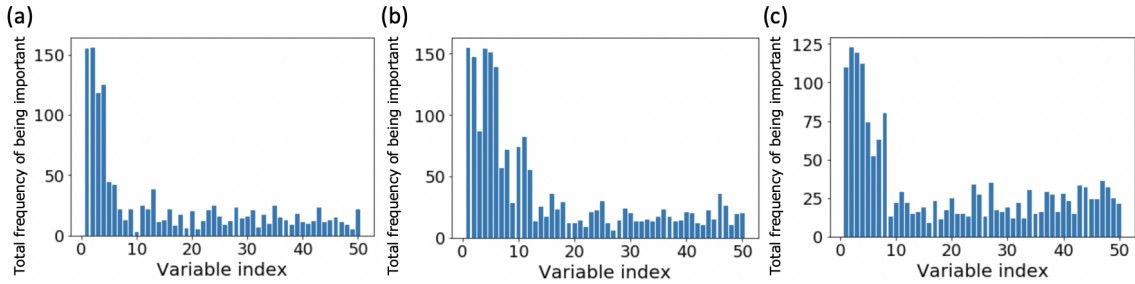

Figure 7: The total frequency of being chosen as important for each variable on the Branin (a), Hartmann6 (b) and Styblinski-Tang4 (c) cases with importance weights $[1, 0.5, 0.1]$.

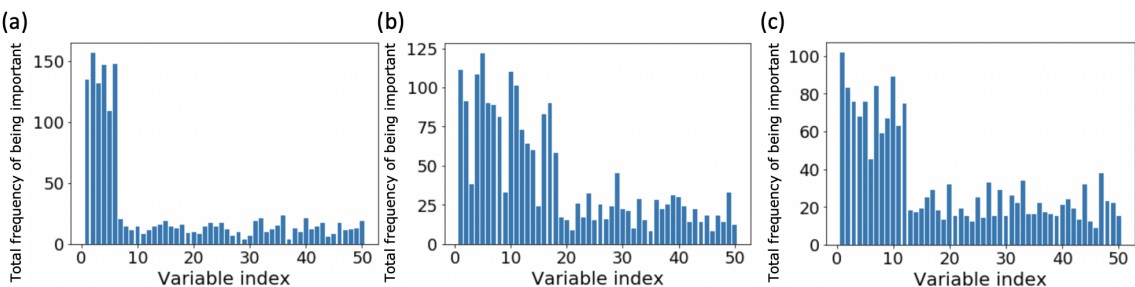

Figure 8: The total frequency of being chosen as important for each variable on the Branin (a), Hartmann6 (b) and Styblinski-Tang4 (c) cases with importance weights $[1, 0.9, 0.8]$.

$[0, 1]^{124}$. It has one objective function $f_{mopta}(\mathbf{x})$ that needs to be minimized and 68 constraints $c_i(\mathbf{x}), i \in \{1, 2, \ldots 68\}$. Similar to Eriksson and Jankowiak (2021), we convert these constraints to soft penalties and convert the minimization problem to the maximization problem by adding a minus at the front of the objective function, i.e., we construct the following new function $F_{mopta}$[9]:

$$F_{mopta}(\mathbf{x}) = -\left(f_{mopta}(\mathbf{x}) + 10 \sum_{i=1}^{68} \max(0, c_i(\mathbf{x}))\right).$$

All experiments for these two real-world problems are run on the same Linux cluster that has 80 2.40 GHz 20-Core Intel Xeon 6148 CPUs.

[9]The Fortran codes of MOPTA08 can be found in https://www.miguelanjos.com/jones-benchmark and we further use codes in https://gist.github.com/denis-bz/c951e3e59fb4d70fd1a52c41c3675187 to wrap it in python.

As described in section 5.2, we design a sampling experiment to test the accuracy of the variable selection in real world problems. The indices of the first 5 variables that have been chosen most frequently are $\{1, 2, 3, 59, 60\}$ in the rover trajectory problem and $\{30, 37, 42, 79, 112\}$ in MOPTA08, and the indices of the first 5 variables that have been chosen least frequently are $\{15, 18, 29, 38, 51\}$ and $\{59, 77, 91, 105, 114\}$ respectively (Figure 2c). The total number of input points in each set is 800000. Figure 2d shows the empirical distributions of function values from two sets of inputs. The significant difference between the two distributions in each panel tells us that changing the values of variables that have been chosen more frequently can alter the function value more significantly, indicating that these variables are more important.

## D  VS-BO with larger numbers of iterations

Figure 9 shows the regret curves for VS-BO over an extended range of iterations, up to 1000. The wall clock times required for 1000 iterations on the Branin, Hartmann6, and Styblinski-Tang4 test cases are approximately 80 minutes, 90 minutes, and 180 minutes, respectively. We notice that there's a large variation in the wall clock time for extended iterations. For instance, the Hartmann6 test case exhibits times ranging from 50 minutes to 120 minutes.

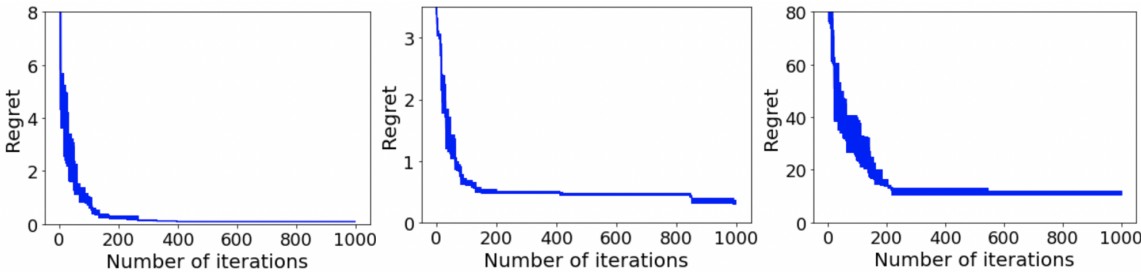

Figure 9: Performance of VS-BO on Branin (left), Hartmann6 (middle) and Styblinski-Tang4 (right) test cases with larger numbers of iterations. For each case, we do 5 independent runs for each method. We plot the mean and standard error of the regret versus iterations.

## E  Sensitive analysis of hyper-parameters of VS-BO

We conduct a study to assess the robustness of two hyper-parameters in VS-BO: $r_{stop}$ and $N_{is}$. The former controls the point at which variable selection stops (line 7 in Algorithm 2), while the latter represents the number of Monte Carlo samples used to estimate the importance score $IS$ (as shown in Equation (5)). We run VS-BO with different $r_{stop}$ values, 5, 10, 50, 100 on the Branin case (Figures 10a left) and the Hartmann6 case (Figures 10a right) and compare their performance. Results show that $r_{stop}$ is highly robust across different values. Likewise, we run VS-BO with different $N_{is}$ values, 1000, 10000, 50000, 100000 on the Branin case (Figures 10b left) and the Hartmann6 case (Figures 10b right). Results show that $N_{is}$ is also highly robust across different values. We choose $r_{stop}$ equal to 10 and choose $N_{is}$ equal to 10000 for all the experiments.

## F  Comparisons of VS-BO, MCTS-VS, and BAxUS

After the preprint of our work (Shen and Kingsford, 2021) was released, Song et al. (2022) published a follow-up work for high-dimensional BO via variable selection; they use Monto Carlo tree search to select variables. Their method is called MCTS-VS. Papenmeier et al. (2022) also proposed a new high-dimensional BO method called BAxUS. Though BAxUS is also an embedding-based algorithm, it differentiates itself from prior embedding-based techniques. Specifically, BAxUS progressively increases the embedding dimension $d$ throughout the BO process, eliminating the necessity to

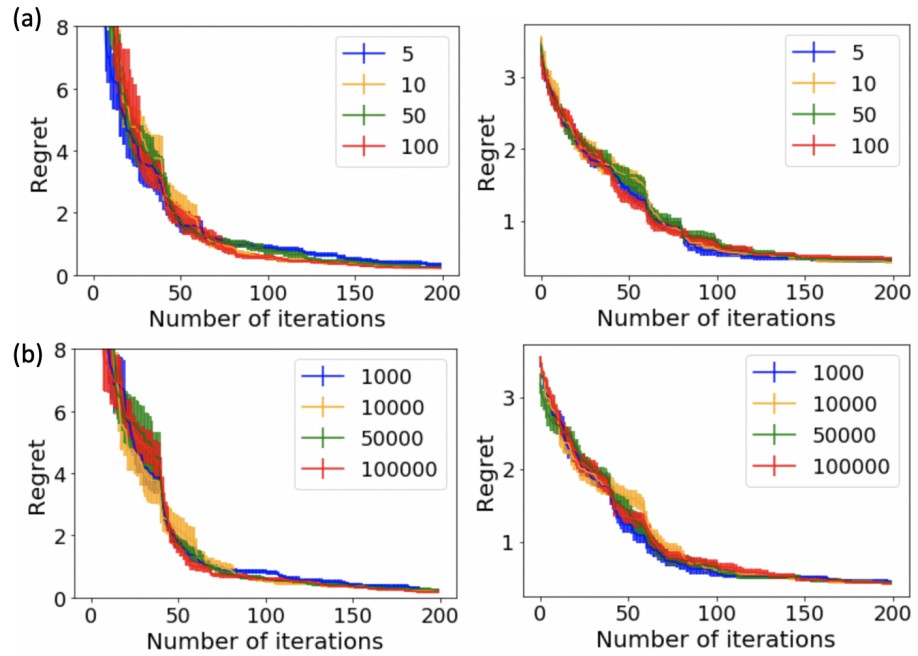

Figure 10: Performance of VS-BO with (a) different values of $r_{stop}$ (5, 10, 50, 100) in both the Branin and Hartmann6 cases (left and right plots, respectively), and (b) different values of $N_{is}$ (1,000, 10,000, 50,000, and 100,000) in the same two cases (left and right plots, respectively).

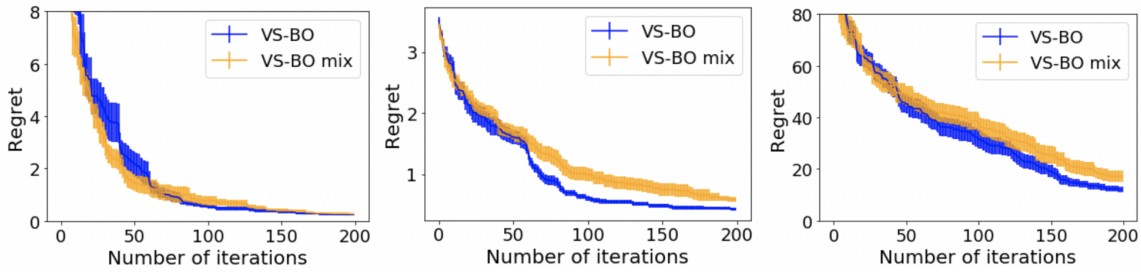

Figure 11: Performance of VS-BO and VS-BO-mix on Branin (left), Hartmann6 (middle) and Styblinski-Tang4 (right) test cases. For each case, we do 20 independent runs for each method. We plot the mean and standard error of the regret versus iterations.

predetermine the value of $d$. Furthermore, to enhance performance, BAxUS is integrated with the local search BO algorithm TuRBO (Eriksson et al., 2019).

We compare VS-BO against MCTS-VS and BAxUS in our three synthetic cases, and Figure 13 shows the results. Similar to other embedding-based methods, BAxUS outperforms both VS-BO and MCTS-VS within a very limited iteration budget, such as the initial 50 iterations. However, BAxUS have more variable performance, in some cases such as Hartmann6 it performs clearly better than other methods, while in other cases such as Styblinski-Tang4 it performs significantly worse. Compared to VS-BO, MCTS-VS seems to need more data to learn an accurate search tree. However, we notice that the wall clock time per iteration in MCTS-VS is substantially shorter than that of VS-BO and BAxUS. As a result, MCTS-VS is preferable for scenarios with expansive iteration budgets (e.g., $\geq 500$ iterations). For tighter iteration constraints, either VS-BO or embedding-based methods might be better. We also notice that the incorporation of local search with new BO algorithms can

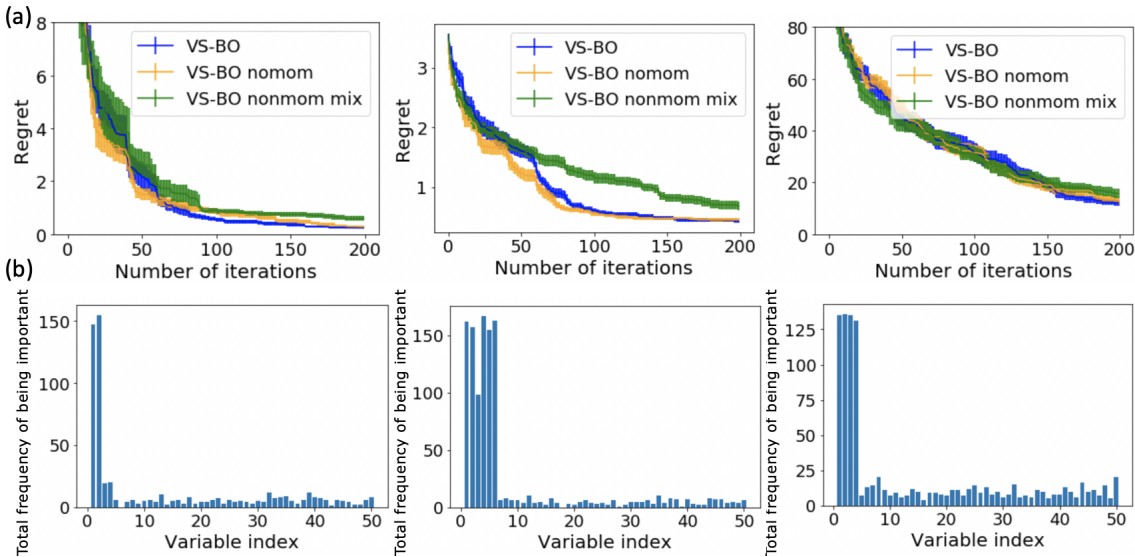

Figure 12: (a) Performance of VS-BO, VS-BO without momentum mechanism (VS-BO-nomom) and VS-BO-nomom with the mixed sampling strategy (VS-BO-nomom-mix) on Branin (left), Hartmann6 (middle) and Styblinski-Tang4 (right) test cases. For each case, we do 20 independent runs for each method. We plot the mean and standard error of the regret versus iterations. (b) The total frequency of being chosen as important for each variable by VS-BO-nomom on Branin (left), Hartmann6 (middle) and Styblinski-Tang4 (right) test cases.

be very helpful. Thus, merging VS-BO with local search strategies like TuRBO (Eriksson et al., 2019) presents an interesting avenue for future exploration.

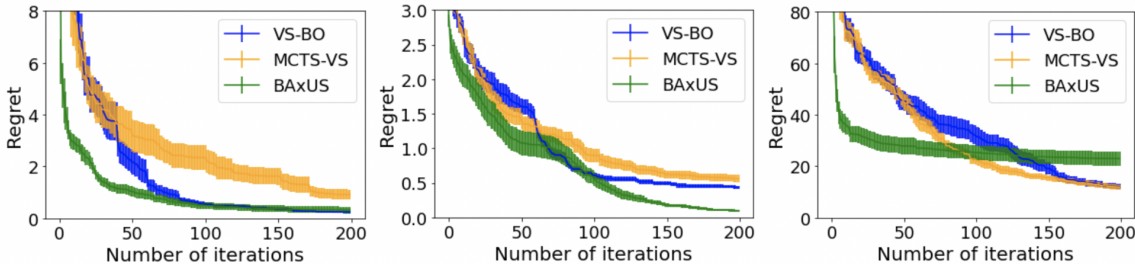

Figure 13: Performance of VS-BO, MCTS-VS and BAxUS on Branin (left), Hartmann6 (middle) and Styblinski-Tang4 (right) test cases. For each case, we do 20 independent runs for each method. We plot the mean and standard error of the regret versus iterations.

