# OpenReview forum: "Computationally Efficient High-Dimensional Bayesian Optimization via Variable Selection"
_automl.cc/AutoML/2023/Conference — AutoML 2023 MainTrack_

### Official Review · Reviewer_cR9i · 2023-03-27

**Potential Impact On The Field Of Automl Rating:** 3
**Technical Quality And Correctness Rating:** 2
**Clarity Rating:** 2

**Summary Of Contributions:**

- The introduction of a novel, adaptive variable selection scheme for high-dimensional Bayesian Optimization
- A novel treatment of important versus unimportant variables through the dual use of BO and CMA-ES
- The BO that results from the proposed variable selection is cheap, both in theory and in practice


**Actions Required To Increase Overall Recommendation:**

I am very open to changing my score, and it greatly hinges on further assessing the quality of the variable selection. In an ideal world, that would entail:
- Showing the results (and frequency plots) of the non-momentum method, preferably with & without CMA-ES (doing RS instead)
- Showing how sensitive the (non-momentum) method is to the hyperparameters
- Ablate at which level (i.e. $0.1f, 0.2f, 0.3f$) variables start to become important for Hartmann-6 and Styb-Tang

Here, I view the first two as must-haves, and the rest being nice-to-haves. I don't expect it to perfectly pick up all relevant dimensions, but there is currently far too many confounding factors.

Further, I would like to see that the computational expense does not blow up (relative to the competition, given the unique (?) scaling factor) when the number of evaluations is increased by a factor $2-5$ - a common high-dimensional BO budget. I recognize that this is unrealistic for ALEBO and SAASBO.

Moreover, the inclusion of the aforementioned NeurIPS works would be nice to have.

**Clarity:**

The algorithm that is used for experiments, VS-momentum, is convoluted and has multiple long, involved algorithms that live mostly in the Appendix. Any steps to simplify this on the reader would be greatly appreciated, but I do not appreciate the fact that the reader _must_ go into the appendix to find the actual algorithm that is used. Please provide intuition into key differences of the momentum algorithm in the method section. To me, the first paragraph and Figure in the Appendix seem important enough to be included in the main paper.

Otherwise, the paper is well-formulated, clear and a pleasant read.

**Overall Review:**

# Positive
- __The relevance is high__: High-dim BO does feature a number of methods which are expensive to the point where even benchmarking against them is painful. Having computationally feasible methods is important, particulairly since the function is likely going to need 100's of evaluations to be optimized, anyway.
- __The idea is good__: Identifying relevant variables and optimizing solely on these is a simple and intuitive idea, which I appreciate. Combining with CMA-ES also seems sensible from a practical perspective.
- __The proposed algorithm is provably cheap__: Given the conventional BO setup, substeps of the algorithm scales computationally with the dimensionality of $x_{ipt}$
- __The evaluation protocol is novel (?) and very practial__: Adding semi- and mini-relevant dimensions to synthetic test functions is a great idea - one which seems substantially more realistic than the current black-and-white regime.
- __Good empirical performance__: Clearly, the method performs well, which is even more impressive given its compute time. However, I _do not_ view this as supremely important.
- __Clear and well-explained__. The paper is mostly a pleasant read, with a high degree of correctness and clarity.


# Negative:
- __Hyperparameters of variable selection__: Algo. 2, Line 7 contains the hyperparameters 3 and 10. These are not mentioned as such, nor ablated. I would greatly appreciate if the authors could shed some light on how they arrived at these values, and provide an ablation on how these impact performance. Moreover, I believe they should be parametrized so that their scalar values are not part of the algorithm.
- __Computational expense & Large number of GPs__: The method involves fitting $D$ GPs, making it scale as $\mathcal{O}(Dn^3)$. To my knowledge, is a uniquely large factor that other methods do not share (rather, they are $\mathcal{O}($D^2 + Dn^2 + n^3)$ as pointed out by the authors. This puts the main claimed benefit of the paper into question, as there should be tasks (pref. those with some multiple on the number of evaluations) where the proposed method is in fact one of the slower ones.
- __Inability to pick up semi-relevant dimensions__: VSBO does not seem to pick up on the semi-important dimensions very well based on the frequency plots. In particular, it struggles with the semi-active dimensions on Styb-Tang, despite the function's extreme steepness along the borders which should simplify matters. Does it pick up any semi-active dimensions (with the current hyperparameter values) for a function with a flatter surface like, say, Hartmann-3? . This could probably be addressed by varying the aforementioned hyperparameters, but I would ultimately want to see that it works as intended out of the box.
- __The algorithm is not digestible__: There is _a lot_ of material, including 5 algorithms totaling over two pages, to go through to get to VS-momentum (i.e. the actual algorithm that is used). To me, this appears to be a design flaw, and _hints_ at over-fitting the algorithm to the problem(s) at hand. Moreover, I want to see the performance of the non-momentum variant.


- __Missing comparison against relevant methods__: Two competing methods from NeurIPS-22, which both work with the (adaptive) axis-aligned subspace assumption, are missing (BaXUS [1],  MCTS-VS [2]). In particular, they are both very relevant with the chosen evaluation protocol. I recognize that these are relatively recent (more than 3 months since publication), but I ultimately do not they are recent enough to warrant exclusion. Since we on the topic of computational efficiency/scalability, TuRBO [3] should probably be in there as well. At the least, these should be included as related work. Having [1] (which uses some of the same benchmarks) and [2] included would improve my perception of the paper.

- __Relation of IS to lengthscales__:The proposed variable selection criterion _seems_ to be very correlated to the lengthscales of the model (except if short lengthscales & little data), and my intuition tells me that the lengthscales should suffice about as well for variable selection. Can the authors provide evidence to the contrary?
- __The MC approximation is not ablated__: The Monte Carlo approximation seems fairly sample-intensive, considering the large dimensionality of the problem. How many samples were used, and does the IS suffer if (relatively) few samples are used?

# General feedback:
- __Improving the motivation for the method__: I greatly appreciate Fig. 4 and what it communicates (although I think a. and c. are unnecessary). I was not aware that the cost of optimizing the GP in high-dim is that significant, and I think it should be highlighted more. In fact, I think it should serve as a motivating example for the entire paper and be featured in the introduction.
- __Excessive time-related plots__: Both wall-clock and CPU time seems extreme, and they are (naturally) just about identical. Please remove wall-clock time to make room for Styb-Tang.
- __Plot the regret__: Please, plot the regret (best possible - current best), especially when you have introduced test new functions where readers may not know the optimal value by heart.
-__CMA-ES blurs the evaluation: The inclusion of CMA-ES blurs the evaluation, especially under the proposed evaluation protocol. In fact, it appears that CMA-ES (and not the variable selection) is responsible for the optimization of the semi-active dimensions in 2/3 synthetic cases, since VSBO doesn't indenify these dimensions.


# Questions
_No, I don't expect experiments, but would greatly appreciate some form of clarity_!
- How susceptible is the proposed variable selection procedure to observation noise of various magnitudes?


# Minor stuff

- I'm missing the boundary issue (Swersky 2016) as a reason for high-dimensional BO being challenging. Would the authors agree?
- Plotting 1/8 standard __deviation__ seems odd. Why not plot the standard __error of the mean__?
- Fig. 4: The y-axis label "Wall clock time __/__ second" reads "Wall clock time __per__ second" and is very is unconventional, please go with "Wall clock time (seconds)"
- __the__ GP is missing almost everywhere, such as in line 73, 74 & 75.
- Algo 5, row 28: You've written () instead of {} in LaTeX.


1. Increasing the Scope as You Learn: Adaptive Bayesian Optimization in Nested Subspaces. Leonard Papenmeier, Luigi Nardi, Matthias Poloczek. In Advances in Neural Information Processing Systems (NeurIPS), 2022.

2. Monte Carlo Tree Search based Variable Selection for High Dimensional Bayesian Optimization. Lei Song, Ke Xue, Xiaobin Huang, Chao Qian. In Advances in Neural Information Processing Systems (NeurIPS), 2022.

3. D. Eriksson, M. Pearce, J. Gardner, R. D. Turner, and M. Poloczek. Scalable Global Optimization via Local Bayesian Optimization. In Advances in Neural Information Processing Systems (NeurIPS), 2019.

**Potential Impact On The Field Of Automl:**

The potential impact on the field of AutoML is high. High-dimensional BO is a relevant topic, and the need for computationally efficient methods is high, given that many of the current best methods (SAASBO, ALEBO) are on the opposite end of the spectrum. With that said, such methods currently exist (see later references), but do not act on the same principles.

**Reproducibility (Optional):**

I have not tried to run the code. Based on the paper, however, it appears as if I would be able to fully re-implement the algorithm(s) based on the level of detail provided.

**Review Confidence:**

5: You are absolutely certain about your assessment. You are very familiar with the related work and checked all the details carefully.

**Review Rating:**

4: Weak Reject: For instance, a paper with minor technical flaws, limited impact, and/or weak evaluation.

**Review Summary:**

The idea of variable selection is good, the evaluation protocol is novel and practical, and the paper could potentially fill a need for computationally efficient high-dimensional BO with consistent performance. However, the algorithm does not appear intrinsically scalable $(\mathcal{O}(Dn^3))$ and it is very difficult to gauge how well the proposed variable selection scheme works, given:
- The use of a modified method, VS-momentum, in the results
- The hyperparameters of the method
- The functions that are benchmarked against

but the total list is even longer. Thus, I am not convinced that the proposed variable selection scheme has merit.

I recommend to reject the paper in its current format. I am confident in my assessment given the current evidence, but I am very open to changing my opinion if my critique is addressed.

**Technical Quality And Correctness:**

The main issue of the paper is that it is almost impossible to assess the effectiveness of the variable selection scheme due to a multitude of design choices and hyperparameters which blur the evaluation. This is greatly expanded on below.

I will note that the work is otherwise technically sound.

---

> ### Author Response · Authors · 2023-05-01
> **Response to the reviewer's comments**
>
> Thank you for reviewing our work, we have revised our manuscript based on your comments. Here are some further responses:
>
> `Hyperparameters of variable selection: Algo. 2, Line 7 contains the hyperparameters 3 and 10. These are not mentioned as such, nor ablated. I would greatly appreciate if the authors could shed some light on how they arrived at these values, and provide an ablation on how these impact performance. Moreover, I believe they should be parametrized so that their scalar values are not part of the algorithm.`
>
> 3 is actually not a hyper-parameter. Since the stopping criterion is defined as the iteration when the difference between successive values of the negative log marginal likelihood is less than a factor of 10 of the difference between successive values at the previous iteration, the number here is at least 3 (we need $L_{m-2}$, $L_{m-1}$, and $L_{m}$).
>
> We have added some new ablation studies about hyper-parameter 10. Please see Figure 9 of the updated manuscript. We find that this hyper-parameter is highly robust across different values.
>
> `Computational expense & Large number of GPs: The method involves fitting...This puts the main claimed benefit of the paper into question, as there should be tasks (pref. those with some multiple on the number of evaluations) where the proposed method is in fact one of the slower ones.`
>
> We think that the reviewers may have misunderstood the computational complexity of our method. Our approach only requires fitting one Gaussian Process (GP) per iteration, which is the same as the standard Bayesian Optimization (BO) framework, instead of D GPs.
>
> Although our method does fit multiple GPs during the variable selection step, two factors mitigate the computational cost. Firstly, assuming that the dimension of $x_{ipt}$, p, is significantly smaller than D, stepwise forward-based variable selection only requires fitting p GPs instead of D GPs. Secondly, we perform variable selection every 20 iterations rather than at each iteration in our experiments.
>
> We acknowledge that in extreme cases where every variable is crucial, the variable selection step may require a longer computation time.
>
> `The MC approximation is not ablated: The Monte Carlo approximation seems fairly sample-intensive, considering the large dimensionality of the problem. How many samples were used, and does the IS suffer if (relatively) few samples are used?`
>
> We use 10000 samples. We have added some new ablation studies about this number. Please see Figure 9 of the updated manuscript.
>
> `Showing the results (and frequency plots) of the non-momentum method, preferably with & without CMA-ES (doing RS instead)`
>
> We have added new results on the comparison among VS-BO, VS-BO-nonmomentum, and VS-BO-nonmomentum without CMA-ES. Please see Figure 11 of the updated manuscript. We find that although the regret curves of VS-BO-nomomentum are highly similar to those of VS-BO, VS-BO-nomomemtum appears to be less effective in selecting secondary important variables, particularly in the Branin case. Therefore, we decide to keep VS-BO (with momentum) in our manuscript.
>
> `Excessive time-related plots: Both wall-clock and CPU time seems extreme, and they are (naturally) just about identical. Please remove wall-clock time to make room for Styb-Tang.`
>
> We have put the Styb-Tang into the main manuscript.
>
> `Plot the regret: Please, plot the regret`
>
> We have plotted the regret for synthetic functions. We don't plot the regret for real-world problems because it is assumed that we don't know the optimal value.
>
> `Plotting 1/8 standard deviation seems odd. Why not plot the standard error of the mean?`
>
> We have used standard error to replace all the standard deviation.
>
> `Fig. 4: The y-axis label "Wall clock time / second" reads "Wall clock time per second" and is very is unconventional, please go with "Wall clock time (seconds)"`
>
> We have revised our manuscript based on this comment.
>
> `the GP is missing almost everywhere, such as in line 73, 74 & 75.`
>
> We have revised our manuscript based on this comment.
>
> `Showing how sensitive the (non-momentum) method is to the hyperparameters`
>
> As we describe above, we have added ablation studies on the variable selection stop criteria and the number of Monte Carlo samplings. Please see Figure 9 of the updated manuscript. We find that both hyper-parameters are highly robust across different values.

---

> > ### Author Response · Authors · 2023-05-01
> > **Response to the reviewer's comments (continue)**
> >
> > `Further, I would like to see that the computational expense does not blow up (relative to the competition, given the unique (?) scaling factor) when the number of evaluations is increased by a factor 2-5`
> >
> > We have found that our method requires approximately 20 minutes to run 500 iterations for the Branin function, 15 minutes for the Hartmann6 function, 35 minutes for the StyblinskiTang4 function, 60 minutes for the Rover function, and 100 minutes for the MOPTA function. Therefore, we consider our method to be practical and feasible for these tasks.
> >
> > We agree that ALEBO or SAASBO cannot handle such a high number of iterations. For example, SAASBO requires approximately 9 hours to complete 200 iterations on the Branin function. We have pointed out this in our updated manuscript.
> >
> > `Relation of IS to lengthscales:The proposed variable selection criterion seems to be very correlated to the lengthscales of the model (except if short lengthscales & little data), and my intuition tells me that the lengthscales should suffice about as well for variable selection. Can the authors provide evidence to the contrary?`
> >
> > Firstly, the lengthscale and variable selection serve different purposes. The lengthscale measures the importance of variables, while variable selection is a method to choose a subset of variables. Hence, the lengthscale cannot replace variable selection.
> >
> > Secondly, we chose not to use the lengthscale for computing the importance score in our work. This is because previous research, specifically "Variable selection for Gaussian processes via sensitivity analysis of the posterior predictive distribution," has demonstrated the disadvantages of the lengthscale and shown that the method we use for computing the importance score is better.
> >
> > `Ablate at which level variables start to become important for Hartmann-6 and Styb-Tang`
> >
> > We have added some new experimental results. We test our method on three new synthetic functions with importance weight [1,0.5,0.1] and [1,0.9,0.8], and the results in Figures 7&8 show that our method can detect correct important variables. Please refer to the updated manuscript for further details.

---

> > > ### Comment · Reviewer_cR9i · 2023-05-01
> > > **Response**
> > >
> > > Thank you for these additional experiments and explanations. I will make sure to give them the proper consideration, and reconsider my rating of the paper.

---

> > > > ### Author Response · Authors · 2023-05-02
> > > > **Response to the reviewer's comments**
> > > >
> > > > Thank you for the reviewer's further response.
> > > >
> > > > We did some new tests on three synthetic functions with maximal iteration 1000. In this situation, we find that our method requires approximately 90 minutes for the Branin function and the Hartmann6 function, and 2 hours for the StyblinskiTang4 function (wall clock time). We also observed that as the maximum iteration increases, the time usage of different runs of the same function exhibits higher variance. For instance, for StyblinskiTang4, the minimum time is 6542 seconds, while the maximum time is 10839 seconds.

---

> > ### Comment · Reviewer_cR9i · 2023-05-01
> > **Further response to rebuttal**
> >
> > Thanks to the reviewers for addressing the concerns regarding the ablations. It is nice to see that the method is indeed robust to the likelihood-related hyperparameter. I appreciate the clarification on $m$, and I consequently agree with the authors. Further, it's nice to see that the "vanilla" (non-momentum, non-CMA-ES) version of the method appears effective. As such, I have increased my score one step. However, I am still not satisfied with regard to the discussion of scalability, which is the central claim of the paper. As a reported strength of the method, it is concerning that experiments are run with so few evaluations (200, TurBO does thousands for comparison) and that it possesses a uniquely large scaling factor. Anecdotal evidence is not sufficient in this regard.
> >
> > ##### Although our method does fit multiple GPs during the variable selection step, two factors mitigate the computational cost. Firstly, assuming that the dimension of $x_{ipt}$, p, is significantly smaller than D, stepwise forward-based variable selection only requires fitting p GPs instead of D GPs.
> >
> > __In Algorithm 2, Line 5 & 6:__
> >
> > __for__ $ m = 1:D$
> >
> > Fit GP to $\mathcal{D_n}$
> >
> > The subsequent if-condition does not guarantee a smaller computational cost.
> >
> > The scaling is clearly $\mathcal{O}(Dn^3)$ as long as the if-statement does not guarantee an early stop - I do not believe I have misunderstood this. To this end, it is unimportant how frequent the variable selection step takes place as long as it's done throughout. I agree that there are mitigating factors which affect the practical computationsl expense, but again - this does not affect the _theoretical_ scalability of the method.
> >
> > ##### We acknowledge that in extreme cases where every variable is crucial, the variable selection step may require a longer computation time.
> > Yes, but $\mathcal{O}(pn^3)$, for some $p < D$ would also be a uniquely large factor that could cause scalability issues.
> >
> >
> > Thanks to the authors for addressing some of my concerns - I am still open to further raising my score if the questions regarding scalability is properly addressed in the experiments.

---

> > > ### Author Response · Authors · 2023-05-01
> > > **Response to the reviewer's comments**
> > >
> > > Thank you for your further response. You are correct that from a theoretical perspective, the scaling for the variable selection step should be $O(Dn^3)$ because we don't provide a theoretical guarantee of the stepwise forward selection (which might be an interesting question we need to think of). We apologize for not being precise enough in our previous response.
> > >
> > > Practically, we have some evidence indicating that the variable selection step is not very time-consuming. First, we added new figures (Figure 6) showing the number of important variables chosen versus iterations. The number of important variables is the same as the number of  GPs we need to fit. We find that they are all less than 10. These results show that we will do less than 10 GPs for each variable selection step.
> > >
> > > Second, as we describe in our responses below, we test our method on all five function cases with max iteration 500. We find that the wall clock time of each case is still manageable and significantly smaller than that of ALEBO and SAASBO. The wall clock time here includes the time to do the variable selection. Therefore, we think our method is practical for larger iterations. We will try to include these results in the manuscript, although it might be challenging due to the page limit.

---

### Review · Reproducibility_Reviewer_Ymh4 · 2023-04-11

**Completeness Of Code And Dataset Supplement Rating:** 3
**Usability And Ease Of Reproducibility Rating:** 2

**Actions Required To Increase The Reproducibility And Overall Recommendation:**

A good option would be to provide a container file, e.g., a singularity definition file that installs all required dependencies and provides entry points to running and plotting the results. It would be good to have some command-line arguments to include certain benchmarks/algorithms in the runs.

Creating a container file ensures the software can be installed on a fresh installation.

See also my other comments.

**Completeness Of Code And Dataset Supplement:**

The authors provide code and dataset descriptions that generally make it possible to reproduce their results. They further provide an archive with their results.

Running the experiments requires some work. There is no general script that would reproduce all experiments. Furthermore, some aspects are missing:

* It is unclear to me how to run the Mopta08 benchmark. The instructions for this are vague. Overall, this seems to be too complicated. There are executables for this benchmark that can be called from Python. I don't see why R would be required here.
* There are no good instructions on installing the methods the authors compare against (such as Alebo, REMBO, SAASBO). The authors could provide these dependencies in the requirements.txt (see, e.g., https://stackoverflow.com/questions/16584552/how-to-state-in-requirements-txt-a-direct-github-source )
* The requirements.txt is not in a proper format, and some package names are wrong (it is "scikit-learn" not "sklearn", and "ax-platform" not "ax"). It would be good to have a requirements.txt that installs *all* necessary dependencies via "pip install -r requirements.txt".
* The Python version is not specified. Some of the packages are in versions incompatible with recent Python versions, such as 3.10 or 3.11. I would expect the Python version to be specified in the README.md
* The "Experiments_script.py" shows how to run the different benchmarks and methods. However, it requires commenting out the benchmarks/methods not to run. It would be better if there was a central script to reproduce all results in order.
* Matplotlib is missing from the list of dependencies. It is required for the plotting functionality.
* The NoiseKernel import is missing from the list of dependencies.
* "psutil" is missing from the list of dependencies
* Instructions to install tmvtnorm are missing.
* CUDA version not working (device mismatch error).

Even though these issues might make it hard for a lay person to reproduce the results, it seems that a sufficiently knowledgeable person would be able to reproduce the results. However, they would need to invest a considerable amount of time.

Generally, the instructions could be

**Overall Reproducibility Review:**

The paper generally provides the code to reproduce the results. However, important instructions for reproducing the results are missing. The folder structure for storing the results is unclear to me. Extracting the provided archive into the results folder caused an error in the plot script.

It would be good if the authors could provide a step-by-step tutorial showing how to install dependencies (under which Python version), run all algorithms on all benchmarks (with the option of leaving out certain runs), and plot the results.

Furthermore, I think it would be good to avoid additional dependencies such as R.

Nevertheless, I believe it is possible to reproduce the results, but only with major efforts. I was able to run VSBO on synthetic benchmarks and got results that were coherent with the plots in the paper.

**Review Confidence:**

4: You are confident in your assessment, but not absolutely certain. It is unlikely, but not impossible, that you did not understand some parts of the submission or that you are unfamiliar with some pieces of the code or data.

**Review Rating:**

6: Borderline: Leaning Accept, all critical aspects are reproducibile with minor effort, and the remainder are likely reproducible with major additional effort.

**Review Summary:**

This paper addresses high-dimensional Bayesian Optimization by finding relevant variables using a gradient approximation and fitting a GP only on the relevant variables (VSBO).

The authors provide code, results, and instructions to reproduce the results. Unfortunately, some parts of the instructions are unclear, making it hard to reproduce the results. I was able to run VSBO on synthetic benchmarks. However, the results could not be stored because folders were missing. Furthermore, I was not able to run the plot script with the folder structure I found to be appropriate.

The authors write that "Monte Carlo Tree Search based Variable Selection for High-Dimensional Bayesian Optimization" (Song et al., 2022) was released after they published a preprint. However, the authors could compare to it for an AutoML '23 submission. It seems to be very relevant. Another high-dimensional BO method from NeurIPS '22 to compare against is "Increasing the Scope as You Learn: Adaptive Bayesian Optimization in Nested Subspaces." (Papenmeier et al., 2022)

**Summary Of Necessary Code And Dataset Supplement:**

This paper addresses the problem of high-dimensional Bayesian Optimization by searching for the important variables for the problem at hand. The important/unimportant variables are assumed to be the variables with a large/small gradient (in magnitude). A Gaussian process surrogate is used to approximate the gradient.

The authors provide an implementation of the method, several test problems, as well as scripts to run the other methods they benchmark against.

The authors mainly implement VSBO in BoTorch. The implementation follows the algorithmic description of the paper and seems to be relatively straightforward to implement, given the pseudocode.

The authors did not create new benchmarks but benchmarked VSBO on several synthetic two well-known real-world benchmarks, the 60-dimensional Rover trajectory planning problem and the 128-dimensional Mopta08 vehicle design problem.

The authors did not re-implement any methods they compared against but used pre-existing implementations.

**Usability And Ease Of Reproducibility:**

I had difficulties reproducing the results:

* Running e.g., Branin gives the following error: "No such file or directory: './benchmark/Branin/2_2_2_50/VSBO/X_1.npy'". It seems that a certain folder structure is expected, but this is not documented in the README.md
* I extracted the raw_results into the "results" folder, so there are folders "Branin", "Hartmann6", etc., inside the 'results' folder. With this structure, I get the following error when trying to run 'plot_script.py': 'FileNotFoundError: [Errno 2] No such file or directory: './VSBO/F_chosen_1.npy'

Further comments:

* Is there no Python alternative to the R package tmvtnorm? It should be possible to sample from a truncated Gaussian in Python.
* It's hard to understand the Experiments_script.py file. It's not clear how to run the different experiments. It would be better to have a separate file for each experiment.
* It is unclear how to run the Mopta experiments.

---

> ### Author Response · Authors · 2023-04-30
> **Response to the reviewer's comments**
>
> Thank you for reviewing our work. We have significantly revised our codes based on your comments. Here are our responses to your comments:
>
> ` It is unclear to me how to run the Mopta08 benchmark. The instructions for this are vague. Overall, this seems to be too complicated. There are executables for this benchmark that can be called from Python. I don't see why R would be required here.`
>
> We also encountered significant difficulties when attempting to compile files for the Mopta08 benchmark during our first attempt. The benchmark is used in the work presented in https://arxiv.org/abs/2103.00349, but no Python codes were provided. In our revised repository, we have included two compiled files, libmopta.so and moptafunc.so, which are available in the mopta_libs.zip file. We believe that simply placing these two files in the same directory as the running script should work, although users may still encounter some missing library issues.
>
> We were not aware until recently that the work "Increasing the Scope as You Learn: Adaptive Bayesian Optimization in Nested Subspaces" provides a Python code for the Mopta benchmark. In the future, we can use their code directly to test our methods.
>
> `There are no good instructions on installing the methods the authors compare against (such as Alebo, REMBO, SAASBO). The authors could provide these dependencies in the requirements.txt (see, e.g., https://stackoverflow.com/questions/16584552/how-to-state-in-requirements-txt-a-direct-github-source )`
>
> We have provided their dependencies in the requirements.txt.
>
> ` The requirements.txt is not in a proper format, and some package names are wrong (it is "scikit-learn" not "sklearn", and "ax-platform" not "ax"). It would be good to have a requirements.txt that installs *all* necessary dependencies via "pip install -r requirements.txt". `
>
> We have revised requirements.txt.
>
> `The Python version is not specified. Some of the packages are in versions incompatible with recent Python versions, such as 3.10 or 3.11. I would expect the Python version to be specified in the README.md`
>
> We use python 3.7, which has been specified in the README.md now.
>
> `The "Experiments_script.py" shows how to run the different benchmarks and methods. However, it requires commenting out the benchmarks/methods not to run. It would be better if there was a central script to reproduce all results in order.`
>
> We have made significant revisions to the running script. Please refer to the updated README.md for instructions on how to run it.
>
> `Matplotlib is missing from the list of dependencies. It is required for the plotting functionality.`
>
> We have added Matplotlib into requirements.txt.
>
> ` The NoiseKernel import is missing from the list of dependencies. `
>
> We have removed NoiseKernel module since it is not used.
>
> `"psutil" is missing from the list of dependencies`
>
> We have removed psutil module since it is not used.
>
> ` Instructions to install tmvtnorm are missing.`
>
> After installing rpy2, there is no need to perform additional Python installations for the tmvtnorm package if it is already installed in R. We have included an introduction to tmvtnorm in our updated README.md and a link on how to install the package in R. Unfortunately, we could not find a Python version of tmvtnorm. For more information, please refer to this answer: https://stackoverflow.com/questions/52489814/python-versions-tmvtnormrtmvnorm-which-original-at-r.
>
> `CUDA version not working (device mismatch error).`
>
> We did not utilize any GPU resources during our experiments, and as such, our code has not been tested on GPU. We have explicitly specified torch.device to be CPU-only.
>
> `Running e.g., Branin gives the following error: "No such file or directory: './benchmark/Branin/2_2_2_50/VSBO/X_1.npy'". It seems that a certain folder structure is expected, but this is not documented in the README.md`
>
> We have revised our repository based on this comment. Please refer to the updated README.md for instructions.
>
> ` I extracted the raw_results into the "results" folder, so there are folders "Branin", "Hartmann6", etc., inside the 'results' folder. With this structure, I get the following error when trying to run 'plot_script.py': 'FileNotFoundError: [Errno 2] No such file or directory: './VSBO/F_chosen_1.npy'`
>
> We have revised our plotting scripts based on this comment. Please refer to the updated README.md for instructions.

---

> > ### Comment · Reproducibility_Reviewer_Ymh4 · 2023-05-01
> > **Response to authors**
> >
> > Thank you for addressing my comments. I think that your changes increase the reproducibility of the paper. Unfortunately, I cannot find an updated version of the code - the anonymous GitHub link in the paper still seems to refer to the old version.
> >
> > I would like to understand how easy it is to reproduce the results after your changes. Can you provide an updated version?

---

> > > ### Author Response · Authors · 2023-05-01
> > > **Response to reviewers**
> > >
> > > That is very weird, because from our side, the anonymous GitHub has already been updated yesterday. Can you try the link again? https://anon-github.automl.cc/r/vsbo-EC08
> > >
> > > Or I can directly send our codes via OpenReview.
> > >
> > > Thank you for your response!

---

> > > > ### Comment · Reproducibility_Reviewer_Ymh4 · 2023-05-01
> > > > **Response to authors**
> > > >
> > > > Thank you for the quick reply. That was my mistake; the code is up-to-date. I'll get back to you later.

---

> > ### Comment · Reproducibility_Reviewer_Ymh4 · 2023-05-02
> > **Response to authors**
> >
> > After revising the code, running VSBO became considerably more straightforward. On Ubuntu 22.04 with Python 3.7.16, "pip install torch==1.13.1+cu117" failed but "torch==1.13.1" worked.
> >
> > I think running the baselines could be simplified by using Ax for all of them. There are strategies for Alebo, HeSBO, and Saasbo implemented in Ax.
> >
> > The code quality could be improved. The `baseline_run` script does not run because the `args = parser.parse_args()` line is missing. Some deprecated code patterns exist, like wildcard imports from VSBO_run and VSBO_utils, while unused imports are in other files. Overall the code structure is difficult to follow. There are several typos in the README.md.
> >
> > The R dependency seems appropriate. I could not find a simple way to sample from a truncated multivariate Gaussian.
> >
> > Overall, the revisions improved reproducibility, and I'm increasing my score to a borderline accept: the basic requirements for reproducibility are just fulfilled, but there is much room for improvement.

---

> > > ### Author Response · Authors · 2023-05-02
> > > **Response to reviewers**
> > >
> > > Thank you for your further response.
> > >
> > > `I think running the baselines could be simplified by using Ax for all of them. There are strategies for Alebo, HeSBO, and Saasbo implemented in Ax.`
> > >
> > > We agree that it is a good strategy. We will try to do that in the future.
> > >
> > > `The baseline_run script does not run because the args = parser.parse_args() line is missing.`
> > >
> > > We have revised  baseline_run.py based on this comment.
> > >
> > > `Some deprecated code patterns exist, like wildcard imports from VSBO_run and VSBO_utils, while unused imports are in other files.`
> > >
> > > When run VSBO_run,py, we do find one userwarning on `gpytorch/lazy/triangular_lazy_tensor.py`, it seems to be the version incompatibility between gpytorch and pytorch. We will try to fix it in the future.
> > >
> > > `Overall the code structure is difficult to follow`
> > >
> > > We will continuously annotate our codes.

---

### Official Review · Reviewer_ctFh · 2023-04-13

**Potential Impact On The Field Of Automl Rating:** 4
**Technical Quality And Correctness Rating:** 3
**Clarity Rating:** 4

**Summary Of Contributions:**

&nbsp;

The authors introduce a new method for high-dimensional Bayesian optimisation based on variable selection.

&nbsp;

**Actions Required To Increase Overall Recommendation:**

&nbsp;

I am willing to increase my score by a further point if the minor issues raised above are addressed.

&nbsp;

**Clarity:**

&nbsp;

In general the paper is very clear and well-written. I have the following minor suggestions which may provide additional benefit to the reader:

&nbsp;

**__MINOR POINTS__**

&nbsp;

1. In Section 1, to illustrate what is meant by important and unimportant variables $x_{ipt}$ and $x_{nipt}$, it may be worth including (space permitting) a graphic such as Figure 1 of Bergstra and Bengio [1] that illustrates why random search can perform better relative to grid search in high-dimensional spaces.

2. Line 68, "a GP". Grammatically, it would be great to include definite/indefinite articles and pluralisation where appropriate e.g. "a GP", "the GP", "GPs".

3. Line 70, It may be worth defining $y^{1:n}$ as $[y^1,...,y^n]^T$ to simplify Equation 2 later on. From the reviewer's (potentially opinionated!) perspective this tends to be the more common means of defining observation vectors in the literature.

4. Line 73, perhaps it would be preferable to refer to $\Theta$ and $\sigma_0$ as hyperparameters of the GP rather than parameters.

5. Line 103, "which is similar to an axis-aligned subspace embedding". Is this not exactly an axis-aligned subspace embedding?

6. Line 111, "hyperparameter in the kernel function".

7. Algorithm 1, line 1, $f(x)$ is given as an input to the algorithm. This may imply that the black-box function is given directly (analytic form) rather than simply the ability to query it.

8. Algorithm 1, line 4, as written, the observations are sampled without noise i.e. $y^i = f(x^i)$.

9. Algorithm 1, line 16, why is $x^{max}$ chosen to be that with maximal $y^i$ as opposed to that with the maximal GP posterior mean?

10. Algorithm 2, line 3, IS[I]. Given that i is used as an index for the data points, it may be worth using a second, distinct index for the dimensions.

11. Algorithm 2, line 4, It may be worth explicitly defining D as the full dimension of the input space.

12. Algorithm 2, line 4, What is the meaning of the subscript s? Is it necessary?

13. Algorithm 1, line 5, $p(x | \mathcal{D})$ is introduced abstractly here while it is later explained that it is chosen to be a multivariate Gaussian distribution. It may be worth explaining that when introducing the algorithm.

14. Line 184, typo, "Because of the properties of Gaussian distributions".

15. Line 503, $alpha_0^2$ and $\rho_{1:D}^2$ are not defined as the kernels signal amplitude and the inverse lengthscales respectively.

16. Line 527, should $K(x_{ipt}...)$ be lower-case k here as a vector quantity? This would be consistent with the main paper.

17. Line 535, in the expression for the UCB acquisition, what is the reason for taking the square root of the dimensionless exploration parameter Beta?

&nbsp;

**Overall Review:**

&nbsp;

**__MAJOR POINTS__**

&nbsp;

1. In the section on related work, the literature on VAE-based embedding methods is relevant but is not described. While I believe it is beyond the scope of the paper to compare against VAE-based embedding methods, references such as [2-4] could be briefly discussed. In terms of the literature on high-D BO, TuRBO [5] could also be discussed.

2. In Algorithm 2, line 7, the stopping criterion is defined as the iteration when the difference between successive values of the negative log marginal likelihood is more less than a factor of 10 of the difference between successive values at the previous iteration. This is a hyperparameter of the algorithm and I am wondering if the authors performed any sensitivity analysis on this stopping criterion?

3. For the experiments, why do the authors report the results with 1/8 standard deviation in place of the standard error? The latter may be preferred if the objective is to establish the relative ranking of algorithms. The former may be preferred if the goal is to showcase the variability in performance of the different algorithms.

4. On line 283, in terms of the claim that VSBO outperforms VSBO-mix, would the errorbars be overlapping in the case that the standard error was plotted instead of 1/8 standard deviations? In other words can this claim be definitely supported? If not, perhaps it should be moderated?

&nbsp;

**__MINOR POINTS__**

&nbsp;

1. On line 2, the authors state that the input domain $\mathcal{X}$ is in $[0, 1]^D$. The algorithm introduced, however, is more broadly applicable for bounded input domains in $\mathbb{R}^D$. As such, it may be worth stating that for the purposes of the paper the authors restrict their study "without loss of generality" to input domains in $[0, 1]^D$.

2. Line 23, "maximizes the function f" assumes a maximization problem. While the intention is understood perhaps "optimization" would be more appropriate before the black-box optimization problem is formerly defined as a maximization problem. This is a very minor nit and one which is a matter of taste!

3. On line 28, the reference Griffiths and Hernández-Lobato was published in Chemical Science [6].

4. Line 49, "Letham et al. also points out that when projecting the optimal point from the embedding space back to the original space, there is no guarantee that the resulting point is within X , and thus the algorithm may fail to find an optimum within the input domain." It is worth mentioning here that this problem can be partially addressed (at least in the case of linear embedding methods such as ALEBO) by mapping the input bounds from the original space to the embedding space. In the case of PCA for example, the embedding-based bounds would take the form of linear inequality constraints.

5. In Equation 1, the mean vector is set to zero. This implicitly assumes that the observations have been standardized and this would be worth mentioning.

6. In Equation 5, the motivation for randomly sampling points $x^{(k)}$ as opposed to grid sampling them, in addition to the expense of grid sampling for high dimensions, is presumably so that computation of the IS is robust to uninformative dimensions as in [1] for example?

7. Moriconi et al. Was published in the machine learning journal.

8. BoTorch was published in NeurIPS 2020.

9. Line 203, typo, "Matérn". This is a major crime against the French language.

10. Line 326, in terms of the applications for computational chemistry it would be particularly exciting to see applications of VS-BO to descriptor-based molecular property prediction and optimization [7].

11. In the references there are some missing capitalizations such as "Bayesian", "Gaussian", "Monte Carlo".

12. Line 577, an appropriate citation for L-BFGS could be included [8].

13. In Figure 6b) are the confidence bands really 1/8 standard deviations? The uncertainty bands would be very large if that is the case!

&nbsp;

**__REFERENCES__**

&nbsp;

[1] Bergstra and Bengio, [Random Search for Hyper-parameter Optimization](https://www.jmlr.org/papers/volume13/bergstra12a/bergstra12a.pdf), JMLR, 2012.

[2] Grosnit et al., [High-Dimensional Bayesian Optimization with Variational Autoencoders and Deep Metric Learning](https://arxiv.org/abs/2106.03609), arXiv, 2021.

[3] Maus et al., [Local latent Space Bayesian Optimization over Structured Inputs](https://proceedings.neurips.cc/paper_files/paper/2022/file/ded98d28f82342a39f371c013dfb3058-Paper-Conference.pdf), NeurIPS 2022.

[4] Notin et al., [Improving black-box optimization in VAE latent space using decoder uncertainty](https://openreview.net/forum?id=F7LYy9FnK2x), NeurIPS 2021.

[5] Eriksson et al., [Scalable Global Optimization via Local Bayesian Optimization](https://proceedings.neurips.cc/paper_files/paper/2019/file/6c990b7aca7bc7058f5e98ea909e924b-Paper.pdf)
, NeurIPS 2019.

[6] Griffiths and Hernández-Lobato, [Constrained Bayesian Optimization for Automatic Chemical Design using Variational Autoencoders](https://pubs.rsc.org/en/content/articlehtml/2019/sc/c9sc04026a), Chemical Science, 2020.

[7] Griffiths et al., [GAUCHE: A Library for Gaussian Processes in Chemistry](https://arxiv.org/abs/2212.04450), arXiv, 2023.

[8] Liu and Nocedal, [On the limited memory BFGS method for large scale optimization](https://link.springer.com/article/10.1007/BF01589116). 330 Math. Program., 45(1-3):503–528, 1989.

&nbsp;

**Potential Impact On The Field Of Automl:**

&nbsp;

High-dimensional black-box optimisation problems are prevalent in a wide range of autoML tasks. The broader field of research is important and the authors' specific contribution is a simple and fast algorithm that has the potential to be the method of choice on a number of problems. The empirical evaluation is also valuable for the community.

&nbsp;

**Reproducibility (Optional):**

&nbsp;

Code is provided in full together with extensive instructions on how to use the package.

&nbsp;

**Review Confidence:**

5: You are absolutely certain about your assessment. You are very familiar with the related work and checked all the details carefully.

**Review Rating:**

9: Strong Accept: Technically flawless paper with major impact and strong evaluation, with no obvious flaws. Should be highlighted in the program.

**Review Summary:**

&nbsp;

A new algorithm is introduced for high-dimensional Bayesian optimization. The algorithm is simple and straightforward to implement, the empirical and theoretical analyses are thorough and the paper is very well-written. As such I recommend acceptance.

&nbsp;

**Technical Quality And Correctness:**

&nbsp;

1. In Equation 2, there is a minus sign in front of the expression for the posterior predictive variance. Is this a typo?

&nbsp;

---

> ### Author Response · Authors · 2023-04-29
> **Response to the reviewer's comments**
>
> Thank you for reviewing our work. Here are our responses to your comments:
>
> `In Equation 2, there is a minus sign in front of the expression for the posterior predictive variance. Is this a typo?`
>
> This is not a typo, it looks weird because we place the term $k(x,x,\Theta)$ as the second term, we have revised its placement to make the notation more clearer.
>
> `Line 68, "a GP". Grammatically, it would be great to include definite/indefinite articles and pluralisation where appropriate e.g. "a GP", "the GP", "GPs".`
>
> We have revised our manuscript based on this comment.
>
> `Line 70, It may be worth defining $y^{1:n}$ as ... to simplify Equation 2 later on. From the reviewer's (potentially opinionated!) perspective this tends to be the more common means of defining observation vectors in the literature.`
>
> We have revised our manuscript based on this comment.
>
> `Line 73, perhaps it would be preferable to refer to \Theta and \sigma_0 as hyperparameters of the GP rather than parameters.`
>
> We have revised our manuscript based on this comment.
>
> `Line 103, "which is similar to an axis-aligned subspace embedding". Is this not exactly an axis-aligned subspace embedding?`
>
> Yes, our method is an axis-aligned subspace embedding, we have revised our manuscript based on this comment.
>
> `Line 111, "hyperparameter in the kernel function".`
>
> We have revised our manuscript based on this comment.
>
> `Algorithm 1, line 1, f(x) is given as an input to the algorithm. This may imply that the black-box function is given directly (analytic form) rather than simply the ability to query it.`
>
> We have revised our manuscript based on this comment.
>
> `Algorithm 1, line 4, as written, the observations are sampled without noise`
>
> We have revised our manuscript based on this comment.
>
> `Algorithm 1, line 16, why is x^{max} chosen to be that with maximal...as opposed to that with the maximal GP posterior mean?`
>
> This is a common choice for Bayesian Optimization, i.e. choose an input with a known high y value, rather than relying solely on a predicted high value.
>
> `Algorithm 2, line 3, IS[I]. Given that i is used as an index for the data points, it may be worth using a second, distinct index for the dimensions.`
>
> We have revised our manuscript based on this comment.
>
> `Algorithm 2, line 4, It may be worth explicitly defining D as the full dimension of the input space.`
>
> We have defined $D$ at the beginning of section 3.
>
> `Algorithm 2, line 4, What is the meaning of the subscript s? Is it necessary?`
>
> Yes it is necessary. We use $x_{s(i)}$ to represent a permutation of variables in the input $x$ that are sorted by their importance scores, if we do not use $x_{s(i)}$, then $x_i$ should represent the i-th variable in $x$ originally.
>
> `Algorithm 1, line 5, p(x|D) is introduced abstractly here while it is later explained that it is chosen to be a multivariate Gaussian distribution. It may be worth explaining that when introducing the algorithm.`
>
> We have revised our manuscript based on this comment.
>
> `Line 184, typo, "Because of the properties of Gaussian distributions".`
>
> We have revised our manuscript based on this comment.
>
> `Line 503, hyper-parameters are not defined as the kernels signal amplitude and the inverse lengthscales respectively.`
>
> We have revised our manuscript based on this comment.
>
> `Line 527, should K() be lower-case k here as a vector quantity? This would be consistent with the main paper.`
>
> Yes, it should be lower-case. We have revised our manuscript based on this comment.
>
> `Line 535, in the expression for the UCB acquisition, what is the reason for taking the square root of the dimensionless exploration parameter Beta?`
>
> There is no specific reason. I think the only reason is to tell people Beta should be non-negative. We use the square root because previous work usually has this UCB formulation, such as https://arxiv.org/pdf/0912.3995.pdf.
>
> `In the section on related work, the literature on VAE-based embedding methods is relevant but is not described. While I believe it is beyond the scope of the paper to compare against VAE-based embedding methods, references such as [2-4] could be briefly discussed. In terms of the literature on high-D BO, TuRBO [5] could also be discussed.`
>
> We have revised our manuscript based on this comment.
>
> `In Algorithm 2, line 7, the stopping criterion is defined as the iteration when the difference between successive values of the negative log marginal likelihood is more less than a factor of 10 of the difference between successive values at the previous iteration. This is a hyperparameter of the algorithm and I am wondering if the authors performed any sensitivity analysis on this stopping criterion?`
>
> We have added ablation studies on hyper-parameters, stopping criterion $r_{stop}$ and the number of Monte Carlo samples $N_{is}$ for estimating the importance score. The results are in section D and Figure 9. We find that these two hyper-parameters are highly robust across different values.

---

> > ### Author Response · Authors · 2023-04-29
> > **Response to the reviewer's comments  (continue)**
> >
> > `For the experiments, why do the authors report the results with 1/8 standard deviation in place of the standard error? The latter may be preferred if the objective is to establish the relative ranking of algorithms. The former may be preferred if the goal is to showcase the variability in performance of the different algorithms.`
> >
> > `On line 283, in terms of the claim that VSBO outperforms VSBO-mix, would the errorbars be overlapping in the case that the standard error was plotted instead of 1/8 standard deviations? In other words, can this claim be definitely supported? If not, perhaps it should be moderated?`
> >
> > We have revised our figures and replaced all the standard deviations with standard errors. When comparing VSBO with VSBO-mix, we find that the errorbars do not overlap in the Hartmann6 case and Styblinski-Tang4 case, so we think that the claim that VSBO outperforms VSBO-mix still holds.
> >
> > `On line 2, the authors state that the input domain...As such, it may be worth stating that for the purposes of the paper the authors restrict their study "without loss of generality" to input domains in [0,1]^{D}`
> >
> > We have revised our manuscript based on this comment.
> >
> > `On line 28, the reference Griffiths and Hernández-Lobato was published in Chemical Science [6].`
> >
> > We have revised our manuscript based on this comment.
> >
> > `Line 49, "Letham et al. also points out that when projecting the optimal point from the embedding space back to the original space, there is no guarantee that the resulting point is within X , and thus the algorithm may fail to find an optimum within the input domain." It is worth mentioning here that this problem can be partially addressed (at least in the case of linear embedding methods such as ALEBO) by mapping the input bounds from the original space to the embedding space. In the case of PCA for example, the embedding-based bounds would take the form of linear inequality constraints.`
> >
> > We have revised our manuscript based on this comment.
> >
> > `In Equation 1, the mean vector is set to zero. This implicitly assumes that the observations have been standardized and this would be worth mentioning.`
> >
> > We have revised our manuscript based on this comment.
> >
> > `In Equation 5, the motivation for randomly sampling points... as opposed to grid sampling them, in addition to the expense of grid sampling for high dimensions, is presumably so that computation of the IS is robust to uninformative dimensions as in [1] for example?`
> >
> > We agree with this opinion, although we didn't point out it explicitly in our manuscript since it is not a major point in our work.
> >
> > `Moriconi et al. Was published in the machine learning journal.`
> >
> > We have revised our manuscript based on this comment.
> >
> > `BoTorch was published in NeurIPS 2020.`
> >
> > We have revised our manuscript based on this comment.
> >
> > `Line 203, typo, "Matérn". This is a major crime against the French language.`
> >
> > We have revised our manuscript based on this comment.
> >
> > `Line 326, in terms of the applications for computational chemistry it would be particularly exciting to see applications of VS-BO to descriptor-based molecular property prediction and optimization [7].`
> >
> > We have cited [7] in the introduction section now.
> >
> > `Line 577, an appropriate citation for L-BFGS could be included [8].`
> >
> > We have cited [8] now.
> >
> > `In Figure 6b) are the confidence bands really 1/8 standard deviations? The uncertainty bands would be very large if that is the case!`
> >
> > We have replaced the standard deviation with the standard error.

---

> > > ### Comment · Reviewer_ctFh · 2023-05-03
> > > **All of my Points have been Addressed**
> > >
> > > All of my points have been addressed and as promised, I increase my score. There would appear to be a typo in Section D of the appendix, "Sensitivity Analysis". I appreciate the additional experiments provided by the authors as well as the clarifications in the author response!

---

### Official Review · Reviewer_YTEk · 2023-04-13

**Potential Impact On The Field Of Automl Rating:** 2
**Technical Quality And Correctness Rating:** 3
**Clarity:** The literature review and contributio…
**Clarity Rating:** 3

**Summary Of Contributions:**

This paper addresses high-dimensional Bayesian optimization problems and proposes a new algorithm that leverages variable selection. The idea is to embed high-dimensional space in low-dimensional space with a novel technique. The paper also analyzes the computational complexity and presents the empirical evaluations with results.

**Actions Required To Increase Overall Recommendation:**

The definitions could be in another section instead of the Background. Especially, since the equations in the background are unexpected, preliminaries would be better. Mathematical notations need a double check as I mentioned some of them.

**Overall Review:**

The proposed algorithm and the idea is simple and clear. The contribution is declared in a good way. However, there are some notations as I mentioned above should be corrected for publishing the paper. Also, in line 128, it is declared that "The algorithm uses different strategies to choose new queries of the variables from two different sets." It is not clear for the rest of the paper.

**Potential Impact On The Field Of Automl:**

The idea proposed in the paper has a small potential as the similar idea for embedding search space proposed in previous works, as declared in the paper.

**Review Confidence:**

3: You are fairly confident in your assessment. It is possible that you did not understand some parts of the submission or that you are unfamiliar with some pieces of related work.

**Review Rating:**

7: Weak Accept: Technically sound paper with moderate-to-high impact and strong evaluation, with perhaps some minor flaws.

**Review Summary:**

Except for minor corrections, the paper is well-written and contains novelties. There are some parts that are not clear enough as I declared above, and my score will be weak accept for consideration of the other reviewer's opinion.

**Technical Quality And Correctness:**

There are some notations that should be corrected, i.e. the numbers representing the sequential should be down instead of up in line 70. It may cause confusion about the root of a number.

---

> ### Author Response · Authors · 2023-04-30
> **Response to the reviewer's comments**
>
> Thank you for reviewing our work, we have revised our manuscript based on your comments. Here are some further responses:
>
> `i.e. the numbers representing the sequential should be down instead of up in line 70. It may cause confusion about the root of a number.`
>
> We use the superscript instead of subscript here because $x_i$ means the i-th entry of the vector $x$. Therefore, we use $x^{i}$ in line 70.
>
> `Also, in line 128, it is declared that "The algorithm uses different strategies to choose new queries of the variables from two different sets." It is not clear for the rest of the paper.`
>
> As we explained in section 3 of our manuscript, we use the Bayesian Optimization framework to choose new queries for important variables, while for unimportant variables, we use CMA-ES to sample their values. We have revised our manuscript to provide further clarity on this point.

---

### Official Review · Reviewer_MVQB · 2023-04-13

**Potential Impact On The Field Of Automl Rating:** 3
**Technical Quality And Correctness Rating:** 3
**Clarity Rating:** 3

**Summary Of Contributions:**

This paper addresses an important problem in the field of Bayesian optimization (BO): high-dimensional BO. They proposed a new computational efficient BO algorithm that leverages variable selection by performing the following steps: (1) select variables that are considered “important”; (2) recommend experiments using GP models fitted using these variables; (3) get observations and adjust variables deemed “important” every few iterations. The method is compared against other SOTA algorithms for high-dimensional BO problems based on both # iterations and wall/CPU times.

**Actions Required To Increase Overall Recommendation:**

I would like to see the author address the questions/concerned raised in the "Technical Quality And Correctness" and "Clarity" sections.

**Clarity:**

Overall the writing of the paper is decent. Section 3.2 can benefit from being more explicit about the algorithm they are using. Section 4, despite being important, seems too straightforward to be included as a whole chapter.

**Overall Review:**

Positive aspects
- This paper is well organized, with clearly stated framework/algorithm and motivation.
- The proposed algorithm is compared against multiple SOTA algorithms. It seems to achieve comparable or better performance on synthetic and real problems and is computationally efficient.

Negative aspects:
- Some parts of the paper can benefit some clarification and critical analysis.
- There might be some room for improvement for the experiments and the definition of the IS score.

**Potential Impact On The Field Of Automl:**

High dimensional BO is of importance to AutoML in performing hyperparameter optimization tasks when the number of hyperparameters is large. Existing algorithms typically rely on lower-dimensional embedding with a pre-specified dimension, and/or can be computationally expensive. This paper is of potential importance as they “choose” the embedding dimension and seems to be computationally efficient compared to many existing algorithms.

**Reproducibility (Optional):**

The codebase can benefit from including essential docstrings and comments. I find it hard to navigate in the repository.

**Review Confidence:**

4: You are confident in your assessment, but not absolutely certain. It is unlikely, but not impossible, that you did not understand some parts of the submission or that you are unfamiliar with some pieces of related work.

**Review Rating:**

8: Accept: Technically sound paper with major impact and strong evaluation, with perhaps some minor flaws.

**Review Summary:**

This is a well-written and thoughtful paper that can potentially have good impact in the BO and AutoML community. There are some issues in the papers but can be fixed through some additional experiments / clarification texts.

**Technical Quality And Correctness:**

Overall the proposed VS-BO framework makes sense to me. The overall workflow of variable selection and Bayesian optimization is clear. The proposed algorithm is tested on multiple synthetic problems and real problems against multiple benchmark algorithms. The algorithm performs well when the progress is plotted versus CPU/wall time, indicating the computational efficiency of the proposed algorithm.

However, I do have several questions, answering which would help me better assess the methodology of the paper.

Variable selection and sampling unimportant variables:
- When computing the importance score (IS) on page 4, have you explored other strategies than sampling uniformly from the design space? It is possible that the sensitivity of the function w.r.t. a feature is high when the function value is also low.
- I find it difficult to have an good interpretation for the distribution p(x | D) which is used to sample unimportant features. When you say D, did you mean just the X’s or the Y’s as well? What happens some features are just purely noise? Could you clarify on that in the paper?
- Desipte having a relatively good understanding of the CMA-ES algorithm, I find it difficult to fully understand what was done. Some pseudo-code/clarification might help.

Experiments:
- I wonder what the performance of the algorithm would look like when (1) in the test problem we have only pure noise in addition to the important features (2) the test problem (e.g., the hartmann problem) has a similar specification as its current form but has weights like [1, 0.9, 0.8] instead of [1, 0.1, 0.01]. I am interested because (1) helps understand the performance of sampling unimportant features better and (2) helps understand the algorithm performance when you would need to sequantially fit more GPs in variable selection. To me these experiments would strengthen the arguments of paper even if it doen’t perform well in some of the situations. To the very least, one should hypothesize what would happen if such problems were tested.
- A plot of |x_ipt| (i.e. number of variables deemed “important”) versus iteration/time would help.
- It seems like SAASBO performs the best when progress is plotted against iterations. Taking a quick look at the repo, the author seems to be using the default setup for the SAASBO algorithm. I am interested in seeing its performance when it’s less computationally expensive (e.g., smaller # samplers, # warm-ups, etc.)

---

> ### Author Response · Authors · 2023-05-01
> **Response to the reviewer's comments**
>
> Thank you for reviewing our work. Here are our responses to your comments:
>
> `When computing the importance score (IS) on page 4, have you explored other strategies than sampling uniformly from the design space? It is possible that the sensitivity of the function w.r.t. a feature is high when the function value is also low.`
>
> In our manuscript, we demonstrate in equations (4)-(5) that computing the Importance Sampling (IS) is not based on the average of function values, but rather on the average of the gradients of the function (scaled by the standard deviation). We acknowledge that it is possible for the sensitivity of a feature to be high when the function value is low, which is why we opt to use the function gradient. Besides, using Monte Carlo sampling to do estimation in (5) seems to be a very natural choice.
>
> `I find it difficult to have an good interpretation for the distribution p(x | D) which is used to sample unimportant features. When you say D, did you mean just the X’s or the Y’s as well? What happens some features are just purely noise? Could you clarify on that in the paper?`
>
> `Despite having a relatively good understanding of the CMA-ES algorithm, I find it difficult to fully understand what was done. Some pseudo-code/clarification might help.`
>
> In our manuscript, we discuss in section 3 and Algorithm 1 that D represents a set of x,y pairs: $D=((x^i,y^i))_{i=1}^{N}$. $p(x|D)$ is actually a learned distribution so that when we sample $x$ from $p(x|D)$, the corresponding $y$ value should be more likely to be higher. In cases where certain features are simply noise, $p(x|D)$ should ideally resemble random sampling for those features.
>
> CMA-ES is an algorithm to learn the distribution $p(x|D)$. One of the advantages of CMA-ES is that the learned distribution $p(x|D)$ is a multivariate Gaussian distribution, which implies that its conditional distribution is also multivariate Gaussian and can be sampled easily.
>
> `I wonder what the performance of the algorithm woud look like when (1) in the test problem we have only pure noise in addition to the important features (2) the test problem (e.g., the hartmann problem) has a similar specification as its current form but has weights like [1, 0.9, 0.8] instead of [1, 0.1, 0.01]. I am interested because (1) helps understand the performance of sampling unimportant features better and (2) helps understand the algorithm performance when you would need to sequantially fit more GPs in variable selection. To me these experiments would strengthen the arguments of paper even if it doen’t perform well in some of the situations. To the very least, one should hypothesize what would happen if such problems were tested.`
>
> We have added some additional experimental results with weights [1,0.5,0.1] and [1,0.9,0.8], and Figure 7&8 in the manuscript shows the total frequency of being chosen as important for each variable in these cases. As shown, our method is able to detect correct important variables in this case.  Please refer to the updated manuscript for further details.
>
> `A plot of |x_ipt| (i.e. number of variables deemd “important”) versus iteration/time would help.`
>
> We have added these experimental results. Please see Figure 6 in the updated manuscript. As shown, only a small number of variables are deemed important in each case, and this number is close to the number of important variables in reality.
>
> `It seems like SAASBO performs the best when progress is plotted against iterations. Taking a quick look at the repo, the author seems to be using the default setup for the SAASBO algorithm. I am interested in seeing its performance when it’s less computationally expensive (e.g., smaller # samplers, # warm-ups, etc.)`
>
> By default, num_warmup=256 and num_samples=256 in SAASBO. We evaluated a modified SAASBO that used num_warmup=128 and num_samples=128. This version of SAASBO ran faster, but was still significantly slower than other methods. SAASBO (num_warmup=256 and num_samples=256) requires approximately 9 hours to complete 200 iterations on the Branin case, SAASBO (num_warmup=128 and num_samples=128) takes around 5.5 hours on the same task, whereas VS-BO can accomplish 500 iterations on this case in around 20 minutes.
>
> We did not incorporate these findings into our updated manuscript because we were unable to complete all of the test cases using the less computation-intensive version of SAASBO. Further decreasing the value of num_warmup or num_samples may speed up SAASBO, but we are not sure if it would affect its performance. We may include these results later if our work is accepted.

---

> > ### Comment · Reviewer_MVQB · 2023-05-08
> > **Response to the authors**
> >
> > Thanks for the additional experiments and clarification.  Most of my concerns have been addressed and I'll raise my score accordingly.

---

### Official Review · Reviewer_3DxR · 2023-04-19

**Potential Impact On The Field Of Automl Rating:** 3
**Technical Quality And Correctness Rating:** 3
**Clarity:** This paper is clear and easy to follow.
**Clarity Rating:** 3

**Summary Of Contributions:**

1. This paper propose a high-dimensional BO algorithms based on variable selection.
2. Experimental results show the superior performance of VS-BO.

**Actions Required To Increase Overall Recommendation:**

More discussions about the application scenarios of the algorithm and more types of real-world experiments under these problems.



**Overall Review:**

Positive
- The proposed method is easy to understand and the paper is easy to follow.

Negative
- The experimental results seem weak. The statistical significance test may be further strengthen the results.
- The types of real-world problems are limited.


**Potential Impact On The Field Of Automl:**

BO is an important research area in AutoML and black-box optimization.
High-dimensional problem is one of the barrier of the application of  BO algorithms.

**Review Confidence:**

4: You are confident in your assessment, but not absolutely certain. It is unlikely, but not impossible, that you did not understand some parts of the submission or that you are unfamiliar with some pieces of related work.

**Review Rating:**

7: Weak Accept: Technically sound paper with moderate-to-high impact and strong evaluation, with perhaps some minor flaws.

**Review Summary:**

This paper is well-written, but the experimental part is weak.

**Technical Quality And Correctness:**

This paper is technical sound.

The computational cost is analyzed.

---

> ### Author Response · Authors · 2023-04-30
> **Response to the reviewer's comments**
>
> Thank you for reviewing our work, we have revised our manuscript based on your comments. Here are some further responses:
>
> `More discussions about the application scenarios of the algorithm`
>
> We have included additional related work on the applications of Bayesian Optimization (BO) in the introduction section. Furthermore, in the Broader Impact Statement, we emphasized that BO can be applied to various scientific fields, including but not limited to machine learning, computational chemistry, and bioinformatics.
>
> `The statistical significance test may further strengthen the results.`
>
> We have updated our manuscript by using mean and standard error instead of mean and standard deviation. This metric can provide a better comparison between different methods.
>
> Additionally, we have included more results in the experimental section of our revised manuscript.

---

### Official Review · Reviewer_tX7y · 2023-04-20

**Potential Impact On The Field Of Automl Rating:** 2
**Technical Quality And Correctness Rating:** 3
**Clarity Rating:** 4

**Summary Of Contributions:**


In this paper, the challenging problem of using a high-dimensional domain in Bayesian optimization is tackled through gaussian process based variable selection. The method presents an alternative to the current state-of-the-art embedding-based approach. The paper shows that the model improves convergence iterations and time when compared with existing approaches. However, as the number of dimensions in input increases, the variable selection approach becomes less accurate.


**Actions Required To Increase Overall Recommendation:**

Following can be improved/addressed,

1. A step-by-step instruction in the readme file would be helpful to run the project for reproducibility. Also, rerunning the code following the instructions will be helpful so that the reviewer will also be able to validate.

2. Specify the primary benefits of the work over using embedding based BO approach given that the authors have suggested using embedding based approach as first preference.

3. Robustness and stability of the variable selection at high dimensions.




**Clarity:**

The paper is well written and brings clarity to even a novice reader on the topic. The paper gives a clear background on Bayesian optimization and also discusses associated techniques such as variable selection, which are currently being used. It also presents a clear distinction between the novel contributions made by the paper. The paper also presents the issues with the proposed approach.

**Overall Review:**

Strength:

1.	The main strength of the approach is that it tries to overcome the sensitivity of embedding dimensions.
2.	The paper presents a novel variable selection approach that can be expanded into other areas.
3.	The paper is well written, with a clear background distinguishing the contribution from the potential downsides.

Weakness:

1.	The paper fails to clarify to the audience the marginal benefits compared with other approaches on high dimensions.
2.	Paper relies on the assumption of separating the important from the non-important, but the robustness and stability


**Potential Impact On The Field Of Automl:**

The potential impact of the paper in the field of autoML will be the novel high dimensionality variable selection approach that has been introduced in the paper. This approach might be able to be expanded to other domains, which will benefit the AutoML community.

**Reproducibility (Optional):**

The code was included for review and a readme file has been attached. However, the instructions are not clear enough to be able to run the complete project independently. Following are the three main issues faced,

1.	Although requirements.txt was attached for package management, there were version conflicts during installation that had to be manually resolved.
2.	Although the authors have linked a github discussion page for compiling Fortran code, the instructions aren’t straightforward to be able to perform it.
3.	The instruction states to run “Experiments_script.py” but when run it fails since func_and_constr is not available.

The reproducibility review was not able to be completed due to the above reasons, mainly (2) and (3).

**Review Confidence:**

3: You are fairly confident in your assessment. It is possible that you did not understand some parts of the submission or that you are unfamiliar with some pieces of related work.

**Review Rating:**

6: Borderline Leaning Accept: Technically sound paper where reasons to accept outweigh reasons to reject. Please use sparingly.

**Review Summary:**


The paper is rated borderline leaning accept due to the following reasons:

1.	Issues in running the code to create reproducible results.
2.	The robustness and stability of the approach cannot be quantified.
3.	Unable to distinguish the benefits of the approach since it was noted that it doesn’t perform as well as the embedding-based approach at high dimensions, which is the goal of the work.


**Technical Quality And Correctness:**


The method relies on the assumption that variables can be divided into important and unimportant variables during variable selection in a Gaussian process. In cases of failed variable selection, the method acts as regular Bayesian optimization, where all the variables are considered. However, the paper failed to present the robustness of this approach and recommendations to the autoML on the stability across different input dimensions.

---

> ### Author Response · Authors · 2023-04-30
> **Response to the reviewer's comments**
>
> Thank you for reviewing our work. Here are our responses to your comments:
>
> `A step-by-step instruction in the readme file would be helpful to run the project for reproducibility. Also, rerunning the code following the instructions will be helpful so that the reviewer will also be able to validate.`
>
> We have significantly revised our codes in the repository. Please refer to the updated README.md for instructions on how to run it.
>
> `Robustness and stability of the variable selection at high dimensions.`
>
> We have added new results to the synthetic experiments by changing the importance weights of the test function from [1, 0.1, 0.01] to [1, 0.5, 0.1] and [1,0.9,0.8] and running our method on the new functions. The results in Figure 7&8 demonstrate that our variable selection step can still detect the actual important variables, which further indicate the robustness of the variable selection module. Please refer to Figure 7&8 for a detailed description of the results.
>
> The real-world experiments have higher dimensions than synthetic experiments, and we actually use a sampling experiment to demonstrate that our variable selection module identifies the actual important variables. Please refer to Section 5.2 and Figure 2(c, d) for a detailed description of the experiment and the results.
>
> All of these results demonstrate the robustness and stability of the variable selection module, particularly at high dimensions.
>
> `Specify the primary benefits of the work over using embedding based BO approach given that the authors have suggested using embedding based approach as first preference.`
>
> We did not suggest using the embedding-based approach as the first preference. In fact, all of our experimental results demonstrate that our method outperforms many embedding-based methods, such as REMBO, ALEBO, and HeSBO, particularly in terms of computational efficiency.
>
> In the conclusion section, we do mention that in some extreme cases, such as when the function has thousands of dimensions, the embedding-based approach may still be the preferred choice because the variable selection step of our method may not be as accurate. However, these cases are extremely rare and difficult to encounter.

---

### Author Response · Authors · 2023-05-02
**acknowledgement to all the reviewers**

We would like to express our sincere gratitude to all the reviewers for their time and effort in reviewing our manuscript and providing valuable feedback. We have carefully considered the feedback provided by the reviewers and revised our manuscript accordingly, particularly in the experimental section. We have also responded to each reviewer individually for addressing the comments and questions.